
# (1,0) gauge theories on the six-sphere

**Usman Naseer[1,2]**

**1** Department of Physics and Astronomy, Uppsala University,
Box 516, SE-751s 20 Uppsala, Sweden
**2** Center for Theoretical Physics, Massachusetts Institute of Technology,
Cambridge, MA 02139, USA

## Abstract

We construct gauge theories with a vector multiplet and hypermultiplets of $(1,0)$ supersymmetry on the six-sphere. The gauge coupling on the sphere depends on the polar angle. This has a natural explanation in terms of the tensor branch of $(1,0)$ theories on the six-sphere. For the vector multiplet we give an off-shell formulation for all supersymmetries. For hypermultiplets we give an off-shell formulation for one supersymmetry. We show that the path integral for the vector multiplet localizes to solutions of the Hermitian-Yang-Mills equation, which is a generalization of the (anti-)self duality condition to higher dimensions. For the hypermultiplet, the path integral localizes to configurations where the field strengths of two complex scalars are related by an almost complex structure.


doi:10.21468/SciPostPhys.6.1.002

# 1  Introduction and summary

The study of supersymmetric theories on curved spaces has led to insights into the dynamics of strongly coupled theories. Several exact results for partition functions and other supersymmetric observables have been obtained using supersymmetric localization. One of the simplest curved spaces that admits global supersymmetries is a $d$-dimensional sphere, $\mathbb{S}^d$. Since $\mathbb{S}^d$ is conformally flat there is a canonical way to put a superconformal field theory (SCFT) on it. For non-conformal theories, radius of $\mathbb{S}^d$ serves as an IR regulator which preserves supersymmetry. Following the seminal work of Pestun [1], theories on $\mathbb{S}^d$ for $d \leq 7$ and with various number of supersymmetries have been studied in [2–7] (see [8] for a review). All known results for supersymmetric partition functions on $\mathbb{S}^d$ can be expressed as analytic functions of $d$ [9]. This analytic continuation is consistent with the decompactification limit of corresponding theories [10]. In [11], partition functions for theories with four and eight supersymmetries were obtained, treating the dimension of $\mathbb{S}^d$ as an analytic parameter.

A generic feature of theories that have been studied using localization is the existence of a Coulomb branch. Exact results remain elusive for theories with minimal supersymmetry in four and six dimensions. For $\mathbb{S}^4$, the localization computation has not been possible due to a technical difficulty [12, 13]: no suitable localization term is known. For $\mathbb{S}^6$ there *was* no known construction with eight supercharges.

The case of supersymmetric theories on $\mathbb{S}^6$ is intriguing. Based on the approach of [14] a limited analysis of the off-shell 6D supergravity was carried out in [15]. It did not find $\mathbb{S}^6$ to be a supersymmetric background. Group theoretic arguments of [16] also seem to suggest that theories on $\mathbb{S}^6$ with eight supersymmetries do not exist: there is no supergroup which contains the isometry group of $\mathbb{S}^6$ and only eight supercharges. An implicit assumption in these arguments is that the bosonic symmetry group of the theory contains the full isometry group of the sphere. However, it is possible that the Lagrangian explicitly breaks part of the sphere isometry. This can be done by adding terms in the Lagrangian that depend on the position on the sphere. Such constructions, though somewhat exotic, are not unfamiliar in supergravity and field theory. From the perspective of supergravity one can have a *supersymmetric* solution

with non-trivial values of background fields that do not necessarily preserve the isometries of the metric. From the perspective of field theory such situations arise by promoting the parameters to be space dependent. For example, maximally supersymmetric theories with varying gauge coupling and theta-angle in four dimensions were considered in [17–19]. Theories with eight supercharges and space dependent coupling on certain four-manifolds were constructed in [20].

Another motivation to look for the construction of $(1,0)$ theories on $\mathbb{S}^6$ is their relation to 6D $(1,0)$ SCFTs [21]. These SCFTs have a tensor branch of vacua on which the low energy theory is the $(1,0)$ super Yang-Mills (SYM). The gauge coupling in the IR theory is related to the VEV of the scalar in the tensor multiplet. Being an SCFT, it can be naturally put on $\mathbb{S}^6$ while preserving all supersymmetries. The IR theory on the tensor branch on $\mathbb{S}^6$ will then be a conventional supersymmetric gauge theory on $\mathbb{S}^6$. The scalar field gets a conformal mass term on $\mathbb{S}^6$ and a constant non-zero value is no longer a solution of equations of motions (EoMs). It is conceivable that the VEV on $\mathbb{S}^6$ gives a supersymmetric theory with non-constant coupling.

Let us demonstrate this point in a little more detail for the case of 4D $\mathcal{N}=4$ SYM. In flat space the theory has a supersymmetric Coulomb branch of vacua parameterized by the VEVs of the scalars. In the notation of [1] the conformally coupled theory on $\mathbb{S}^4$ of radius $r$ is

$$\mathcal{L} = \tfrac{1}{2}F_{MN}F^{MN} - \Psi \slashed{D}\Psi + \tfrac{2}{r^2}\phi_I \phi^I, \quad \delta A_M = (\epsilon \Gamma_M \Psi), \quad \delta \Psi = \tfrac{1}{2}F_{MN}\Gamma^{MN}\epsilon + \tfrac{1}{2}\phi_I \Gamma^{\mu I}\nabla_\mu \epsilon. \quad (1.1)$$

Let us look for a supersymmetric solution where $A_\mu = \Psi = [\phi_I, \phi_J] = 0$. Such solution has to satisfy

$$\nabla^2 \phi^I = \tfrac{2}{r^2}\phi^I, \quad \nabla_\mu \phi_I \Gamma^{\mu I}\epsilon + \tfrac{1}{2}\phi_I \Gamma^{\mu I}\nabla_\mu \epsilon = 0. \quad (1.2)$$

Clearly a non-zero constant value of the scalar fields does not satisfy the above constraints. The simplest solution to the EoM is $\phi^I = c^I \left(1 + \beta^2 x^2\right)$, where $c^I$ is constant, $\beta = \tfrac{1}{2r}$, and $x^\mu$ are stereographic coordinates on $\mathbb{S}^4$. This solution breaks half of the 32 supersymmetries. Broken supersymmetries correspond to special conformal supersymmetries in the flat space while the Poincaré supersymmetries are preserved. This analysis is closely related to the existence of half-BPS scalar field configurations of maximally supersymmetric theories on AdS spacetime [22].

Let us now turn to 6D theories. The scalar in the tensor multiplet couples to the vector multiplet in the following way

$$\int \phi \operatorname{Tr}(F \wedge \star F) + \tfrac{1}{2}\partial_\mu \phi \partial^\mu \phi + \cdots, \quad (1.3)$$

where ... denote terms irrelevant to our discussion. On the tensor branch this interaction gives rise the effective gauge coupling of the theory

$$\frac{1}{g_{\text{YM}}^2} = \langle \phi \rangle. \quad (1.4)$$

On $\mathbb{S}^6$, however one has a conformal mass term $\tfrac{3}{r^2}\phi^2$ for the scalar field in (1.3). In its presence a constant $\phi$ with other fields vanishing is not a solution of the EoMs. The simplest solution is $\phi \propto \left(1 + \beta^2 x^2\right)^2$. The effective theory on the tensor branch on $\mathbb{S}^6$ has a position dependent coupling. It remains to be shown that such a solution preserves some amount of supersymmetry on the sphere.

We shall see in this paper that the heuristic picture presented above emerges from a more rigorous analysis. In this paper we construct theories on $\mathbb{S}^6$ with features described above, i.e., a non-constant coupling and $(1,0)$ supersymmetry. We first construct Lagrangian for the $(1,0)$ vector multiplet. We start from a flat-space theory and then deform the Lagrangian

and the supersymmetry transformations to obtain sufficient conditions to put this theory on a curved space while preserving supersymmetry. Indeed we find that by allowing the coupling to depend on the polar angle, we can construct a supersymmetric theory on $\mathbb{S}^6$. We find the following profile for the effective coupling.

$$\frac{1}{g_{\text{eff}}^2} = \frac{1}{g_{\text{YM}}^2}\left(1 + \beta^2 x^2\right)^2, \tag{1.5}$$

where $g_{\text{YM}}$ is a constant parameter with the dimension of length. The coupling is zero at the south pole ($x \to \infty$) and smoothly varies to a non-zero value $g_{\text{YM}}$ at the north pole ($x \to 0$). The position dependence of the coupling is the same as argued above. The resulting Lagrangian is invariant only under an $SO(6) \subset SO(7)$ isometry group of $\mathbb{S}^6$, which leaves the polar angle fixed. We then carry out a similar analysis for hypermultiplets and construct a supersymmetric Lagrangian on $\mathbb{S}^6$.

With an eye towards application of localization we give an off-shell formulation of these theories. For the vector multiplet, we start from an off-shell formulation on $\mathbb{R}^6$ for all supersymmetries and obtain an off-shell formulation on $\mathbb{S}^6$ for all supersymmetries. For hypermultiplets, however, we give an off-shell formulation only for a particular supercharge. This involves introducing pure spinor-like objects as is familiar from off-shell formulation of higher dimensional SYM [23].

Using localization, we show that the path integral for the vector multiplet localizes onto solutions of the Hermitian Yang-Mills (HYM) equation on $\mathbb{S}^6$ which is a generalization of the (anti-)self-duality condition on gauge field in 4D. Solutions of the HYM equation correspond to extended non-perturbative configurations in 6D. The path integral for the hypermultiplet localizes onto configurations where the 1-form field strengths of two complex scalars of the hypermultiplet are related via an almost complex structure. In the perturbative sector (i.e., vanishing gauge field) a simple solution of the localization locus is also a solution of the EoMs of the classical action. This solution, however, diverges at the south pole which leads to a divergent classical action.

The rest of this paper is organized as follows: In section 2 we start by introducing necessary notation and our approach in 4D. We then explicitly construct the action and off-shell supersymmetry transformations for the (1,0) vector multiplet in section 3. In section 4 we do the same for interacting hypermultplets. In section 5 we apply the localization procedure to these theories and obtain the localization locus. We present our conclusions and discuss further issues in section 6. Appendices contain our conventions and technical details of numerous computations.

## 2 Warm-up in four dimensions

Before considering $\mathbb{S}^6$, it is instructive to do the analysis in a familiar four-dimensional setting. Curved four-manifolds admitting supersymmetric gauge theories have been studied in detail using supergravity techniques (see for example [24–26]). Our goal here is simple. We start from a theory on $\mathbb{R}^4$ and modify the Lagrangian and supersymmetry transformations explicitly to put it on a curved manifold $\mathcal{M}^4$.

One can not impose a real structure on the minimal complex super-Poincare algebra in Euclidean 4D. So the construction of a minimally supersymmetric theory requires one to double the number of degrees of freedom (DoFs). Formally, this can be done by considering a field and its hermitian conjugate as transforming independently under the supersymmetry transformations. The path integral over the bosonic fields is understood as a choice of a half-dimensional contour in the space of complex fields. The path integral over the fermionic fields is an al-

gebraic operation defined by the rules of Berezin integration. With this understanding, the Lagrangian for the minimally supersymmetric theory on $\mathbb{R}^4$ is[1]

$$\mathcal{L}_{\mathbb{R}^4} = \tfrac{1}{2}F^2 + \tfrac{1}{2}\psi^1 \not{\partial} \psi^2 - \tfrac{1}{2}\psi^2 \not{\partial} \psi^1 + \tfrac{1}{2}D^2, \tag{2.1}$$

where $D$ is an auxiliary field. The supersymmetry transformations are given by

$$
\begin{aligned}
\delta A_\mu &= \left(\xi^1 \, \Gamma_\mu \psi^2\right) - \left(\xi^2 \Gamma_\mu \psi^1\right), & \delta D &= -\left(\xi^1 \not{\partial} \psi^2 + \xi^2 \not{\partial} \psi^1\right), \\
\delta \psi^1 &= -\tfrac{1}{2}F_{\mu\nu}\Gamma^{\mu\nu}\xi^1 + D\xi^1, & \delta \psi^2 &= -\tfrac{1}{2}F_{\mu\nu}\Gamma^{\mu\nu}\xi^2 - D\xi^2.
\end{aligned}
\tag{2.2}
$$

In Minkowskian signature $\psi^1(\xi^1)$ is related to $\psi^2(\xi^2)$ by complex conjugation but in Euclidean signature they are *a priori* independent. $\psi^1(\xi^1)$ have positive chirality while $\psi^2(\xi^2)$ have negative chirality.

Let us introduce the following useful notation,

$$\psi^i \equiv \begin{pmatrix} \psi^1 \\ \psi^2 \end{pmatrix}, \qquad \xi^i \equiv \begin{pmatrix} \xi^1 \\ \xi^2 \end{pmatrix}. \tag{2.3}$$

The indices $i, j, \ldots$ take values 1 and 2. They are raised and lowered by the antisymmetric matrix $\varepsilon_{ij}$ defined as

$$\varepsilon_{ij} \equiv \begin{pmatrix} 0 & 1 \\ -1 & 0 \end{pmatrix}, \qquad \varepsilon^{ij} \equiv \begin{pmatrix} 0 & 1 \\ -1 & 0 \end{pmatrix}, \qquad \varepsilon_{ij}\varepsilon^{ik} = \delta_i{}^k. \tag{2.4}$$

The auxiliary field can also be expressed as the $2 \times 2$ matrix

$$D^{ij} = -D \begin{pmatrix} 0 & 1 \\ 1 & 0 \end{pmatrix}. \tag{2.5}$$

With this notation, the Lagrangian and the supersymmetry tranformations can be written in a compact form. To put the theory on a curved four-manifold $\mathcal{M}^4$ we start by covariantizing the Lagrangian and the supersymmetry transformations.

$$\mathcal{L}_{\mathcal{M}^4} = \tfrac{1}{2}F^2 + \tfrac{1}{2}\psi^i \not{\nabla}\psi^j \, \varepsilon_{ij} - \tfrac{1}{4}D^{ij}D_{ij},$$
$$\delta A_\mu = \left(\xi^i \Gamma_\mu \psi^j\right)\varepsilon_{ij}, \qquad \delta\psi^i = -\tfrac{1}{2}F_{\mu\nu}\Gamma^{\mu\nu}\xi^i + D^{ij}\xi_j, \qquad \delta D^{ij} = 2\xi^{(i}\not{\nabla}\psi^{j)}. \tag{2.6}$$

The change in $\mathcal{L}_{\mathcal{M}^4}$ under a supersymmetry transformation is (see appendix B.1 for derivation)

$$\delta\mathcal{L}_{\mathcal{M}^4} = \tfrac{1}{2}F_{\mu\nu}\left(\psi_i\Gamma^\rho\Gamma^{\mu\nu}\nabla_\rho\xi^i\right). \tag{2.7}$$

The action of two consecutive supersymmetry transformations on fields is given by (see appendix B.2)

$$
\begin{aligned}
\delta^2 A_\mu &= \mathcal{L}_\nu A_\mu - \nabla_\mu (A_\nu \nu^\nu), \\
\delta^2 \psi^i &= \nu^\mu \nabla_\mu \psi^i + \Gamma^{\mu\nu}\xi^i \left(\psi_j \Gamma_\nu \nabla_\mu \xi^j\right), \\
\delta^2 D^{ij} &= \nu^\mu \nabla_\mu D^{ij} + 2\left(\xi^{(i}\not{\nabla}\xi_k\right)D^{j)k} - F_{\mu\nu}\left(\xi^{(i}\Gamma^\rho\Gamma^{\mu\nu}\nabla_\rho\xi^{j)}\right),
\end{aligned}
\tag{2.8}
$$

where $\nu^\mu$ is the vector field $\nu^\mu \equiv \xi^i \Gamma^\mu \xi_i$. The first term in $\delta^2 A_\mu$ is a Lie derivative along the vector $\nu^\mu$ while the second term is a gauge transformation w.r.t parameter $-A_\nu \nu^\nu$. To complete the construction, we need to show that the r.h.s of eq. (2.7) vanishes up to total derivatives and $\delta^2$ generates a bosonic symmetry of the theory.

---

[1]We drop the overall factor if inverse coupling squared for notational simplicity throughout this paper.

There are two simple ways to satisfy both of these conditions. One is to assume that the supersymmetry parameter is a Killing spinor (KS)

$$\nabla_\mu \xi^i = 0. \tag{2.9}$$

This ensures that $\delta \mathcal{L}_{\mathcal{M}^4} = 0$ and $\nu^\mu$ is a covariantly constant vector. All but the first term in $\delta^2 \psi^i$ and $\delta^2 D^{ij}$ vanish and the supersymmetry algebra indeed closes on a bosonic symmetry. The integrability condition implies that for a non vanishing solution of the KS equation one must have $R_{\mu\nu} = 0$. This gives a way of putting minimally supersymmetric theories on Ricci flat spaces.

Another way to satisfy both conditions is to let the supersymmetry parameter be a conformal Killing spinor (CKS), in which case

$$\nabla_\mu \xi^i = \Gamma_\mu \tilde{\xi}^i. \tag{2.10}$$

$\delta \mathcal{L}_{\mathcal{M}^4}$ and the last term in $\delta^2 D^{ij}$ vanish due to a numerical accident, i.e., $\Gamma^\rho \Gamma_{\mu\nu} \Gamma_\rho = 0$ in four dimensions. The second term in $\delta^2 D^{ij}$ does not vanish or take the form of a bosonic symmetry for arbitrary $\tilde{\xi}^i$. However, for $\tilde{\xi}^i \propto \xi_i{}^2$,

$$\xi^{(i} \slashed{\nabla} \xi_k D^{j)k} = 4\xi^{(i} \tilde{\xi}_k D^{j)k} = 0. \tag{2.11}$$

Using the Fierz identity in eq. (A.10) one can explicitly show that the second term in $\delta^2 \psi^i$ becomes

$$\Gamma^{\mu\nu} \xi^i \left( \psi_j \Gamma_\nu \nabla_\mu \xi^j \right) = \tfrac{1}{2} \left( \tilde{\xi}^j \Gamma_{\mu\nu} \xi_j \right) \Gamma^{\mu\nu} \psi^i = \tfrac{1}{4} \nabla_\mu \nu_\nu \Gamma^{\mu\nu} \psi^i. \tag{2.12}$$

This along with the first term forms the spinorial Lie derivative along the vector field $\nu^\mu$. The condition $\tilde{\xi}^i \propto \xi_i$ also ensures that $\nu^\mu$ is a Killing vector field and hence the supersymmetry algebra closes onto bosonic symmetries of the theory.

Since $\mathbb{S}^4$ admits non trivial CKS, we conclude that a minimally supersymmetric theory can be defined on $\mathbb{S}^4$. The Lagrangian we constructed is the same as the one found in [12] by the dimensional reduction of 5D SYM.

# 3 (1,0) vector multiplet on $\mathbb{S}^6$

We now perform an analysis, similar to that of the previous section, in 6D. We will focus on putting the (1,0) theory on $\mathbb{S}^6$. We shall see that the situation is different in this case as a mere covariantization does not give a supersymmetric theory on $\mathbb{S}^6$. One has to modify the Lagrangian and the supersymmetry transformations further.

## 3.1 Lagrangian and supersymmetry transformations

On-shell (1,0) vector multiplet consists of a gauge field and a Weyl-fermion with four DoFs each. Off-shell, they have five and eight DoFs respectively. For off-shell supersymmetry we need three auxiliary fields. They fit nicely with the notation already introduced, where we impose

$$D^{ij} = D^{ji}, \qquad i, j = 1, 2. \tag{3.1}$$

but the components are otherwise independent[3]. The flat-space Lagrangian and the supersymmetry transformations have the same form as in eq. (2.6) with an important difference:

---

[2]The proportionality constant is $\beta$ for $\mathbb{S}^4$ and is fixed by the integrability condition.

[3]Usually, one would use three real auxiliary fields $K_1, K_2, K_3$ in terms of which the auxiliary field contribution is manifestly positive-definite ( proportional to $K_1^2 + K_2^2 + K_3^2$ ).

$\psi^1$, $\psi^2$, $\xi^1$ and $\xi^2$ have the same chirality. The field content contains eight fermionic DoFs of the same chirality and supersymmetry transformations are generated by eight parameters of the same chirality. The Lagrangian and the supersymmetry transformations depend on these parameters holomorphically and the theory is chiral.

To put the theory on a curved manifold $\mathcal{M}^6$, we proceed as in the case of four dimensions by covariantizing the flat-space theory. The variation of the Lagrangian under a supersymmetry transformation and two supersymmetry transformations of the fields are given by

$$
\begin{aligned}
\delta \mathcal{L}^{\text{vec}}_{\mathcal{M}^6} &= \tfrac{1}{2} F_{\mu\nu} \left( \psi_i \Gamma^\rho \Gamma^{\mu\nu} \nabla_\rho \xi^i \right) + \text{t.d}, \\
\delta^2 A_\mu &= \mathcal{L}_\nu A_\mu - \nabla_\mu (A_\nu \nu^\nu), \qquad \delta^2 \psi^i = \nu^\mu \nabla_\mu \psi^i + \Gamma^{\mu\nu} \xi^i \left( \psi_j \Gamma_\nu \nabla_\mu \xi^j \right), \\
\delta^2 D^{ij} &= \nu^\mu \nabla_\mu D^{ij} + 2 \left( \xi^{(i} \slashed{\nabla} \xi_k \right) D^{j)k} - \tfrac{1}{2} F_{\mu\nu} \nabla_\rho \left( \xi^{(i} \Gamma^{\rho\mu\nu} \xi^{j)} \right) - F_{\mu\nu} \left( \xi^{(i} \Gamma^\nu \nabla^\mu \xi^{j)} \right),
\end{aligned}
\tag{3.2}
$$

where t.d is a total derivative term

$$
\text{t.d} = -\tfrac{1}{4} \nabla_\rho \left( F_{\mu\nu} \left( \psi_i \Gamma^{\rho\mu\nu} \xi^i \right) \right) + \tfrac{1}{2} \nabla_\mu \left( F^{\mu\nu} \left( \psi_i \Gamma_\nu \xi^i \right) \right) - \tfrac{1}{2} \nabla_\rho \left( D^{ij} \left( \xi_i \Gamma^\rho \psi_j \right) \right).
\tag{3.3}
$$

For a KS, the supersymmetry algebra closes off-shell and the Lagrangian is invariant. In this way we can construct supersymmetric theories on CY3-folds as was done in [15]. For a CKS, $\delta \mathcal{L}_{\mathcal{M}^6}$ does not vanish. This is essentially because $\Gamma^\rho \Gamma_{\mu\nu} \Gamma_\rho \neq 0$ in 6D. A modification of the covariantized Lagrangian and (or) supersymmetry transformations is required to construct a supersymmetric theory.

To proceed, we specialize to $\mathbb{S}^6$ and choose a set of CKSs. The round metric on $\mathbb{S}^6$ is

$$
ds^2_{\mathbb{S}^6} = \frac{1}{(1 + \beta^2 x^2)^2} dx^{\hat{\mu}} dx^{\hat{\mu}} = e^{\hat{\mu}} e^{\hat{\mu}},
\tag{3.4}
$$

where indices with a hat are the *flat* indices. Indices on coordinates and their differentials will always be flat indices, i.e., $x^\mu = x^{\hat{\mu}} = x_\mu = x_{\hat{\mu}}$. A set of frame fields is defined by $e^{\hat{\mu}}$ as

$$
e^{\hat{\mu}} \equiv e^{\hat{\mu}}{}_\mu dx^\mu \equiv \frac{1}{1 + \beta^2 x^2} dx^\mu.
\tag{3.5}
$$

We choose spinor parameters $\xi^i$ to be

$$
\xi^i = \frac{1}{\sqrt{1 + \beta^2 x^2}} \epsilon^i,
\tag{3.6}
$$

where $\epsilon^i$ is a constant spinor. This is a particular set of solutions of CKS equation with[4]

$$
\tilde{\xi}^i = \partial_\mu f \, \Gamma^\mu \xi^i, \quad f \equiv -\tfrac{1}{2} \log \left( 1 + \beta^2 x^2 \right).
\tag{3.7}
$$

We take the Lagrangian on $\mathbb{S}^6$ to be the covariantized Lagrangian multiplied with a factor of $e^\phi$, where $\phi$ is a position dependent function on $\mathbb{S}^6$. Using the supersymmetry variation in eq. (3.2) and the total derivative term eq. (3.3) we get

$$
\delta \mathcal{L}^{\text{vec}}_{\mathbb{S}^6} = e^\phi \left( \partial_\rho f + \tfrac{1}{4} \partial_\rho \phi \right) \left[ F_{\mu\nu} \left( \psi_i \Gamma^{\rho\mu\nu} \xi^i \right) - 2 F_{\rho\nu} \left( \psi_i \Gamma^\nu \xi^i \right) \right] + \tfrac{1}{2} e^\phi \partial_\rho \phi \, D^{ij} \left( \psi_i \Gamma^\rho \xi_j \right).
\tag{3.8}
$$

If we choose $\phi = -4f$, then

$$
\delta \mathcal{L}^{\text{vec}}_{\mathbb{S}^6} = \tfrac{1}{3} e^\phi \, D^{ij} \left( \slashed{\nabla} \xi_i \psi_j \right).
\tag{3.9}
$$

---

[4]The other set is given by $\frac{\Gamma^{\hat{\mu}} x^{\hat{\mu}}}{\sqrt{1 + \beta^2 x^2}} \epsilon^i$. In $r \to \infty$ limit this corresponds to conformal supersymmetries of flat space.

We can get rid of this last term by modifying the supersymmetry transformation of the auxiliary fields to

$$\delta' D^{ij} = \delta D^{ij} + \tfrac{2}{3}\left(\slashed{\nabla}\xi^{(i}\psi^{j)}\right).$$ (3.10)

This gives a Lagrangian on $\mathbb{S}^6$ which is invariant under eight supersymmetry transformations generated by eight independent solutions of the CKS equation given in eq. (3.6). The overall factor of $e^{\phi}$ captures the position dependence of the inverse squared coupling and it matches with the heuristic argument given in the introduction.

### 3.2 Closure of the supersymmetry algebra

We now compute the action of two supersymmetry transformations on fields in our construction of (1,0) vector multiplet on $\mathbb{S}^6$. For covariantized transformations, this is already given in eq. (3.2). The supersymmetry transformation of the auxiliary field is modified from the covariantized version as in eq. (3.10). Only the variation of the gaugino and auxiliary fields is affected by this modification.

#### 3.2.1 The gaugino

For the gaugino we get

$$\delta'^2\psi^i = \delta^2\psi^i + \tfrac{2}{3}\left(\psi^{(i}\slashed{\nabla}\xi^{j)}\right)\xi_j.$$ (3.11)

We do the computation explicitly for $i = 1$. After using eq. (B.29) and CKSE we get

$$\delta'^2\psi^1 = v^\mu\nabla_\mu\psi^1 - \Gamma_{\mu\nu}\xi^1\left(\xi^j\Gamma^\rho\Gamma^{\mu\nu}\psi_j\right)\partial_\rho f + 2\left(\psi^1\Gamma^\mu\xi^j\right)\xi_j\partial_\rho f + 2\left(\psi^j\Gamma^\mu\xi^1\right)\xi_j\partial_\mu f.$$ (3.12)

Using the second Fierz identity in eq. (A.13) we can write

$$\Gamma_{\mu\nu}\xi^1\left(\xi^j\Gamma^\rho\Gamma^{\mu\nu}\psi_j\right)\partial_\rho f = \frac{\partial_\rho f}{4}\left(\xi^1\Gamma^\sigma\xi^j\right)\Gamma_{\mu\nu}\Gamma^\sigma\Gamma^\rho\Gamma^{\mu\nu}\psi_j + \frac{\partial_\rho f}{48}\left(\xi^1\Gamma^{\sigma\delta\gamma}\xi^j\right)\Gamma_{\mu\nu}\Gamma_{\sigma\delta\gamma}\Gamma^\rho\Gamma^{\mu\nu}\psi_j.$$ (3.13)

After simplifying the gamma matrix structures appearing on the r.h.s[5] and expanding the contraction over the $j$ index we get

$$\Gamma_{\mu\nu}\xi^1\left(\xi^j\Gamma^\rho\Gamma^{\mu\nu}\psi_j\right)\partial_\rho f = -\tfrac{1}{4}v_\mu\partial_\nu f\Gamma^{\mu\nu}\psi^1 + \tfrac{15}{4}v^\mu\partial_\mu f\psi^1 - \tfrac{1}{4}\partial_\mu f\left(\xi^1\Gamma^{\mu\nu\rho}\xi^2\right)\Gamma_{\nu\rho}\psi^1 \\ + \tfrac{1}{4}\partial_\mu f\left(\xi^1\Gamma^{\mu\nu\rho}\xi^1\right)\Gamma_{\nu\rho}\psi^2.$$ (3.14)

The third term in eq. (3.12) can be written in the following way by using the second identity in eq. (A.13).

$$2\partial_\mu f\xi_j\left(\psi^1\Gamma^\mu\xi^j\right) = \tfrac{1}{2}\partial_\mu f\left(\xi_j\Gamma_\nu\xi^j\right)\Gamma^\nu\Gamma^\mu\psi^1 + \tfrac{1}{24}\partial_\mu f\left(\xi_j\Gamma_{\nu\rho\sigma}\xi^j\right)\Gamma^{\nu\rho\sigma}\Gamma^\mu\psi^1 \\ = -\tfrac{1}{2}v_\mu\partial_\nu f\Gamma^{\mu\nu}\psi^1 - \tfrac{1}{2}v^\mu\partial_\mu f\psi^1.$$ (3.15)

Similarly, using Fierz identity the fourth term in eq. (3.12) becomes

$$2\partial_\mu f\xi_j\left(\psi^j\Gamma^\mu\xi^1\right) = -\tfrac{1}{2}\partial_\mu f\left(\xi^j\Gamma_\nu\xi^1\right)\Gamma^\nu\Gamma^\mu\psi_j - \tfrac{1}{24}\partial_\mu f\left(\xi^j\Gamma_{\nu\rho\sigma}\xi^1\right)\Gamma^{\nu\rho\sigma}\Gamma^\mu\psi_j \\ = \tfrac{1}{2}\partial_\mu f\left(\xi^j\Gamma_\nu\xi^1\right)\Gamma^{\mu\nu}\psi_j - \tfrac{1}{2}\partial_\mu f\left(\xi^j\Gamma^\mu\xi^1\right)\psi_j - \tfrac{1}{4}\partial_\mu f\left(\xi^j\Gamma^{\mu\nu\rho}\xi^1\right)\Gamma_{\nu\rho}\psi_j.$$ (3.16)

Expanding the contraction over $j$ index we get

$$2\partial_\mu f\xi_j\left(\psi^j\Gamma^\mu\xi^1\right) = -\tfrac{1}{4}v_\mu\partial_\nu f\Gamma^{\mu\nu}\psi^1 - \tfrac{1}{4}v^\mu\partial_\mu f\psi^1 \\ - \tfrac{1}{4}\partial_\mu f\left(\xi^1\Gamma^{\mu\nu\rho}\xi^1\right)\Gamma_{\nu\rho}\psi^2 + \tfrac{1}{4}\partial_\mu f\left(\xi^1\Gamma^{\mu\nu\rho}\xi^2\right)\Gamma_{\nu\rho}\psi^1.$$ (3.17)

---

[5]We use mathematica package FeynCalc [27,28] for gamma matrix manipulations

Finally, combining all the terms we see that

$$
\begin{aligned}
\delta'^2 \psi^1 &= v^\mu \nabla_\mu \psi^1 - v_\mu \partial_\nu f \, \Gamma^{\mu\nu} \psi^1 + 3 v^\mu \partial_\mu f \, \psi^1 \\
&= v^\mu \nabla_\mu \psi^1 + \tfrac{1}{4} \nabla_\mu v_\nu \Gamma^{\mu\nu} \psi^1 + 3 v^\mu \partial_\mu f \, \psi^1 .
\end{aligned}
\tag{3.18}
$$

A similar result holds for $\psi^2$ with $1 \to 2$ in the above equation.

### 3.2.2 Auxiliary fields

We have

$$
\begin{aligned}
\delta'^2 D^{ij} &= \delta^2 D^{ij} - \tfrac{1}{6} F_{\mu\nu} \left[ \left( \xi^i \Gamma^{\nu\mu} \slashed{\nabla} \xi^i \right) + (i \leftrightarrow j) \right] + \tfrac{1}{3} \left[ D^{jk} \left( \xi_k \slashed{\nabla} \xi^i \right) + (i \leftrightarrow j) \right] \\
&= v^\mu \nabla_\mu D^{ij} + \left[ \left( \xi^i \slashed{\nabla} \xi_k \right) D^{jk} + (i \leftrightarrow j) \right] + \tfrac{1}{3} \left[ D^{jk} \left( \xi_k \slashed{\nabla} \xi^i \right) + (i \leftrightarrow j) \right] \\
&\quad - \tfrac{1}{2} F_{\mu\nu} \left[ \left( \xi^i \Gamma^\rho \Gamma^{\mu\nu} \nabla_\rho \xi^j \right) + (i \leftrightarrow j) \right] - \tfrac{1}{6} F_{\mu\nu} \left[ \left( \xi^i \Gamma^{\nu\mu} \slashed{\nabla} \xi^i \right) + (i \leftrightarrow j) \right] .
\end{aligned}
\tag{3.19}
$$

We now simplify above terms using CKS equation and Clifford algebra commutation relations. For terms involving the auxiliary fields we get

$$
\begin{aligned}
\left( \xi^i \slashed{\nabla} \xi_k \right) D^{jk} + (i \leftrightarrow j) &= 3 v^\mu \partial_\mu f \, \varepsilon^i{}_k D^{jk} + (i \leftrightarrow j) = 6 v^\mu \partial_\mu f \, D^{ij} . \\
D^{jk} \left( \xi_k \slashed{\nabla} \xi^i \right) + (i \leftrightarrow j) &= v^\mu \partial_\mu f \, \varepsilon_k{}^i D^{jk} + (i \leftrightarrow j) = -2 v^\mu \partial_\mu f \, D^{ij} .
\end{aligned}
\tag{3.20}
$$

For terms involving the gauge field we have

$$
\begin{aligned}
-\tfrac{1}{2} F_{\mu\nu} \left[ \left( \xi^i \Gamma^\rho \Gamma^{\mu\nu} \nabla_\rho \xi^j \right) + (i \leftrightarrow j) \right] &= -2 F_{\mu\nu} \partial_\rho f \left( \xi^i \Gamma^{\mu\nu\rho} \xi^j \right) , \\
-\tfrac{1}{6} F_{\mu\nu} \left[ \left( \xi^i \Gamma^{\nu\mu} \slashed{\nabla} \xi^i \right) + (i \leftrightarrow j) \right] &= -2 F_{\mu\nu} \partial_\rho f \left( \xi^j \Gamma^{\nu\mu\rho} \xi^j \right) .
\end{aligned}
\tag{3.21}
$$

These two terms precisely cancel each other. So we conclude that

$$
\delta'^2 D^{ij} = v^\mu \partial_\mu D^{ij} + 4 v^\mu \partial_\mu f \, D^{ij} .
\tag{3.22}
$$

### 3.2.3 The closure

The non-trivial part of two supersymmetry variations (i.e., the part without gauge transformation term) of a field $\Phi$ is

$$
\delta^2 \Phi = \mathcal{L}_v + \Omega_\Phi \left( 2 v^\mu \partial_\mu f \right) \Phi \equiv \delta_v \Phi ,
\tag{3.23}
$$

where the second term is a Weyl transformation with weight $\Omega_\Phi$. We have $\Omega_A = 0, \Omega_\psi = \frac{3}{2}$, $\Omega_D = 2$. We now show that the action is invariant under the bosonic transformation $\delta_v$. We note that $v^\mu$ is a conformal Killing vector (CKV). Using the CKS equation and the gamma matrix identities, we find

$$
\nabla_\mu v_\nu = 4 \partial_{[\mu} f \, v_{\nu]} + 2 v^\rho \partial_\rho f \, g_{\mu\nu} .
\tag{3.24}
$$

Moreover $v^\mu$ is actually constant,

$$
v^\mu = \xi^i \Gamma^\mu \xi_i = \frac{e^\mu{}_{\hat\mu}}{1 + \beta^2 x^2} \epsilon^i \Gamma^{\hat\mu} \epsilon_i = \delta^\mu{}_{\hat\mu} \epsilon^i \Gamma^{\hat\mu} \epsilon_i ,
\tag{3.25}
$$

The transformation $\delta_v$ acts only on dynamical fields and leaves the background fields invariant. In particular

$$
\delta_v g_{\mu\nu} = \mathcal{L}_v g_{\mu\nu} + \delta_{\text{Weyl}} g_{\mu\nu} = 0 .
\tag{3.26}
$$

The Lie derivative of metric along $v^\mu$ is

$$
\mathcal{L}_v g_{\mu\nu} = 4 v^\rho \partial_\rho f \, g_{\mu\nu} .
\tag{3.27}
$$

Then the Weyl transformation of the metric is fixed to be

$$\delta_{\text{Weyl}}\, g_{\mu\nu} = -4\nu^\rho \partial_\rho f\; g_{\mu\nu}. \tag{3.28}$$

We now examine the variation of the action $S = \int d^6 x \sqrt{g}\, \mathcal{L}^{\text{vec}}_{\mathbb{S}^6}$ under $\delta_\nu$. We first consider the action of the Lie derivative. The factor $\sqrt{g}\, e^\phi$ transforms as a scalar density of weight $\frac{2}{3}$, i.e.,

$$\mathcal{L}_\nu \sqrt{g}\, e^\phi = \nu^\mu \partial_\mu \big(\sqrt{g}\, e^\phi\big) + \frac{2}{3}\sqrt{g}\, e^\phi\, \partial_\mu \nu^\mu. \tag{3.29}$$

Since $\partial_\mu \nu^\mu = 0$, this terms essentially transforms as a scalar. Since all indices are contracted properly the rest of the terms in the Lagrangian also transform as a scalar, hence

$$\mathcal{L}_\nu S \;=\; \int d^6 x\, \nu^\mu \partial_\mu \big(\sqrt{g}\, \mathcal{L}^{\text{vec}}_{\mathbb{S}^6}\big) = \int d^6 x\, \partial_\mu \big(\nu^\mu \sqrt{g}\, \mathcal{L}^{\text{vec}}_{\mathbb{S}^6}\big) = 0. \tag{3.30}$$

We next look at the action of the Weyl transformations with respect to a parameter $\Omega = 2\nu^\mu \partial_\mu f$. It is easier to work with a finite version of the infinitesimal Weyl transformation appearing in $\delta_\nu$. Under a Weyl transformation $g_{\mu\nu} \to e^{2\Omega} g_{\mu\nu}$ and $\Phi \to e^{-\Omega_\Phi \Omega}\Phi$. The Weyl transformations of different terms in the Lagrangian are

$$\begin{aligned}
g^{\mu\mu'} g^{\nu\nu'} F_{\mu\nu} F_{\mu'\nu'} &\to e^{-4\Omega} g^{\mu\mu'} g^{\nu\nu'} F_{\mu\nu} F_{\mu'\nu'}, \\
\psi^i \slashed{\nabla} \psi_i &\to e^{-4\Omega}\big(\psi^i \slashed{\nabla}\psi_i + \partial_\mu \Omega \psi^i \Gamma^\mu \psi_i\big), \\
D^{ij} D_{ij} &\to e^{-4\Omega} D^{ij} D_{ij}, \\
\sqrt{g}\, e^\phi &\to e^{4\Omega} \sqrt{g}\, e^\phi.
\end{aligned} \tag{3.31}$$

Note that the second term in the Weyl transformation for the fermion kinetic term is trivially zero. So the Weyl transformation leaves the action invariant. We have shown that the supersymmetry algebra closes off-shell for the supersymmetry parameter $\xi^i$. This conclusion is true more generally when one considers the anti-commutator of two supersymmetry variations w.r.t two different parameters $\xi^i$ and $\zeta^i$. In that case $\nu^\mu = \xi^i \Gamma^\mu \zeta_i$. The supersymmetry algebra here is isomorphic to the $(1,0)$ superPoincaré algebra in $\mathbb{R}^6$.

We have only focused on an abelian gauge group but all the analysis goes though for any gauge group by merely gauge-covariantizing different terms appearing in the action and supersymmetry transformations. To summarize, the Lagrangian[6]

$$\mathcal{L}^{\text{vec}}_{\mathbb{S}^6} \;=\; \frac{e^\phi}{g^2_{\text{YM}}}\Big[\tfrac{1}{2}F^2 + \tfrac{1}{2}\psi^i \slashed{D}\psi^j\, \varepsilon_{ij} - \tfrac{1}{4}D^{ij} D_{ij}\Big], \tag{3.32}$$

is invariant under supersymmetry transformations

$$\delta A_\mu = \big(\xi^i \Gamma_\mu \psi^j\big)\varepsilon_{ij}, \quad \delta\psi^i = -\tfrac{1}{2}F_{\mu\nu}\Gamma^{\mu\nu}\xi^i + D^{ij}\xi_j, \quad \delta D^{ij} = 2\xi^{(i}\slashed{D}\psi^{j)} + \tfrac{2}{3}\big(\psi^{(i}\slashed{D}\xi^{j)}\big), \tag{3.33}$$

where $D_\mu \equiv \nabla_\mu + A_\mu$ is the gauge-covariant derivative. We assume that the gauge indices are contracted using an invariant bilinear form. We also assume the real form of the gauge group so that generators are anti-hermitian. This completes the construction of $(1,0)$ vector multiplet on $\mathbb{S}^6$. One may wonder that the divergence of the overall factor might make the action divergent for generic field configurations[7] . However, one can scale all the fields by a factor of $g_{\text{YM}} e^{-\phi}$. This eliminates the overall factor and the Lagrangian in terms of new fields is manifestly regular at all points on the sphere.

---

[6]Omitting the trace over gauge indices.

[7]We thank B. Assel for raising this point.

# 4  (1,0) hypermultiplet on $\mathbb{S}^6$

We now construct the Lagrangian for (1,0) hypermultiplet on $\mathbb{S}^6$. We start by considering free, on-shell hypermultiplet in $\mathbb{R}^6$ and then systematically modify the Lagrangian and supersymmetry transformations to obtain an interacting, off-shell hypermultiplet on $\mathbb{S}^6$.

## 4.1  Free hypermultiplet

The (1,0) hypermultiplet in 6D consists of four bosonic and fermionic degrees of freedom on-shell. We denote scalars by $2 \times 2$ matrices $\phi^{i\bar{i}}$ and the fermions as $\chi^{\bar{i}}$. The fermion $\chi^{\bar{i}}$ has opposite chirality as compared to the supersymmetry parameter $\xi^i$. Both barred and unbarred indices are separately raised and lowered by $2 \times 2$ antisymmetric matrices. The flat-space Lagrangian

$$\mathcal{L}_{\mathbb{R}^6}^{\text{hyp}} = \tfrac{1}{2}\partial_\mu \phi^{i\bar{j}}\partial^\mu \phi_{i\bar{j}} + \tfrac{1}{4}\chi^i \slashed{\partial}\chi_i, \tag{4.1}$$

is invariant under supersymmetry transformations

$$\delta\phi^{i\bar{j}} = \xi^i \chi^{\bar{j}}, \quad \delta\chi^{\bar{i}} = 2\slashed{\nabla}\phi^{j\bar{i}}\xi_j. \tag{4.2}$$

On $\mathbb{S}^6$ the covariantized version of above Lagrangian and supersymmetry transformations gives

$$\delta\mathcal{L}_{\mathbb{S}^6}^{\text{hyp}} = \nabla_\mu \phi^{i\bar{j}}\left(\chi_{\bar{j}}\Gamma^\rho \Gamma^\mu \nabla_\rho \xi_i\right) = -4\nabla_\mu \phi^{i\bar{j}}\left(\chi_{\bar{j}}\Gamma^\mu \tilde{\xi}_i\right), \tag{4.3}$$

which does not vanish in general. We modify the supersymmetry variation of the fermion to

$$\delta'\chi^{\bar{i}} = \delta\chi^{\bar{i}} + c\,\phi^{j\bar{i}}\slashed{\nabla}\xi_j. \tag{4.4}$$

Under modified supersymmetry transformations

$$\delta'\mathcal{L}_{\mathbb{S}^6}^{\text{hyp}} = (3c - 4)\nabla_\mu \phi^{i\bar{j}}\left(\chi_{\bar{j}}\Gamma^\mu \tilde{\xi}_i\right) - \tfrac{9}{2}c\phi^{i\bar{j}}\left(\chi_{\bar{j}}\xi_i\right). \tag{4.5}$$

First term vanishes if $c = \tfrac{4}{3}$. The second term can be cancelled by adding a mass term for scalars with a specific coefficient. It can be checked easily that the right term is $\tfrac{3}{r^2}\phi^{i\bar{j}}\phi_{i\bar{j}}$, i.e., the conformal mass term in 6D.

## 4.2  Coupling to a vector multiplet

To include interactions with the gauge field we replace the derivatives with the gauge-covariant derivatives in the appropriate representation of the gauge group. Supersymmetry requires additional terms coupling the hypermultiplet to other fields in the vector multiplet. The minimal terms which couple the gaugino $\psi^i$ and the auxiliary field $D_{ij}$ to the hypermultiplet are

$$\psi_i \phi^{i\bar{j}}\chi_{\bar{j}}, \qquad D_{ij}\phi^i_{\ \bar{k}}\phi^{j\bar{k}}, \tag{4.6}$$

where we have suppressed the gauge group indices. We claim that the supersymmetric coupling of the hypermultiplet to the vector multiplet is given by the following Lagrangian.

$$\mathcal{L}_{\mathbb{S}^6}^{\text{hyp}} = \tfrac{1}{2}D_\mu \phi^{i\bar{j}}D^\mu \phi_{i\bar{j}} + \tfrac{1}{4}\chi^i \slashed{D}\chi_i + \tfrac{3}{r^2}\phi^{i\bar{j}}\phi_{i\bar{j}} + \psi_i \phi^{i\bar{j}}\chi_{\bar{j}} - \tfrac{1}{2}D_{ij}\phi^i_{\ \bar{k}}\phi^{j\bar{k}}. \tag{4.7}$$

To prove our claim we explicitly compute the variation of the above Lagrangian under supersymmetry transformations. Due to the coupling with the gauge field, the variation of the first three terms is no longer zero.

$$\begin{aligned}
\delta\Big(\tfrac{1}{2}D_\mu \phi^{i\bar{j}}D^\mu \phi_{i\bar{j}} + \tfrac{1}{4}\chi^i \slashed{D}\chi_i &+ \tfrac{3}{r^2}\phi^{i\bar{j}}\phi_{i\bar{j}}\Big) \\
&= \tfrac{1}{2}F_{\mu\nu}\phi_{i\bar{j}}\left(\chi^{\bar{j}}\Gamma^{\mu\nu}\xi^i\right) - \tfrac{1}{4}\left(\xi^i \Gamma_\mu \psi_i\right)\left(\chi_{\bar{i}}\Gamma^\mu \chi^{\bar{i}}\right).
\end{aligned} \tag{4.8}$$

The variation of the fourth term in $\mathcal{L}_{\mathbb{S}^6}^{\text{hyp}}$ gives

$$\delta\left(\psi_i\phi^{i\bar{j}}\chi_{\bar{j}}\right) = -\tfrac{1}{2}F_{\mu\nu}\phi_{i\bar{j}}\left(\chi^{\bar{j}}\Gamma^{\mu\nu}\xi^i\right) - D_{ij}\phi^{i\bar{k}}\left(\xi^j\chi_{\bar{k}}\right) + \left(\chi_{\bar{j}}\psi_i\right)\left(\xi^i\chi^{\bar{j}}\right)$$
$$- 2D_\mu\phi^i{}_{\bar{j}}\left(\psi_k\Gamma^\mu\xi_i\right)\phi^{k\bar{j}} - 8\left(\psi_i\tilde{\xi}_k\right)\phi^{i\bar{j}}\phi^k{}_{\bar{j}}. \tag{4.9}$$

The third term in above equation can be modified using the Fierz identity on the second line of eq. (A.13).

$$\left(\chi_{\bar{j}}\psi_i\right)\left(\xi^i\chi^{\bar{j}}\right) = \tfrac{1}{4}\left(\xi^i\Gamma_\mu\psi_i\right)\left(\chi_{\bar{i}}\Gamma^\mu\chi^{\bar{i}}\right) - \tfrac{1}{48}\left(\xi^i\Gamma_{\mu\nu\rho}\psi_i\right)\left(\chi_{\bar{i}}\Gamma^{\mu\nu\rho}\chi^{\bar{i}}\right) = \tfrac{1}{4}\left(\xi^i\Gamma_\mu\psi_i\right)\left(\chi_{\bar{i}}\Gamma^\mu\chi^{\bar{i}}\right). \tag{4.10}$$

Last equality follows due to the fact that $\psi_i$ is antisymmetric in the gauge indices but the bilinear $\left(\chi_{\bar{i}}\Gamma^{\mu\nu\rho}\chi^{\bar{i}}\right)$ is symmetric. Hence we see that the combined supersymmetry variation of the first four terms in $\mathcal{L}_{\mathbb{S}^6}^{\text{hyp}}$ is

$$-D_{ij}\phi^{i\bar{k}}\left(\xi^j\chi_{\bar{k}}\right) - 2D_\mu\phi^i{}_{\bar{j}}\left(\psi_k\Gamma^\mu\xi_i\right)\phi^{k\bar{j}} - 8\left(\psi_i\tilde{\xi}_k\right)\phi^{i\bar{j}}\phi^k{}_{\bar{j}}. \tag{4.11}$$

The supersymmetry variation of the term involving the coupling of auxiliary field with hypermultiplet bosons can similarly be computed.

$$\tfrac{1}{2}\delta\left(D_{ij}\phi^i{}_{\bar{k}}\phi^{j\bar{k}}\right) = -8\left(\psi_i\tilde{\xi}_k\right)\phi^{i\bar{j}}\phi^k{}_{\bar{j}} - 2D_\mu\phi^i{}_{\bar{j}}\left(\psi_k\Gamma^\mu\xi_i\right)\phi^{k\bar{j}} - D_{ij}\phi^{i\bar{k}}\left(\xi^j\chi_{\bar{k}}\right). \tag{4.12}$$

First two terms come from variation of $D_{ij}$ and by doing an integration by parts to remove derivatives acting on the gaugino. Last term comes from the variation of hypermultiplet scalars. This cancels the supersymmetry variation of the first four terms and we conclude that the Lagrangian $\mathcal{L}_{\mathbb{S}^6}^{\text{hyp}}$ given in eq. (4.7) is invariant under supersymmetry transformations.

Notice that we did not use the explicit form of the CKS in above argument. The hypermultiplet can be coupled to a background vector multiplet while preserving all sixteen conformal Killing supersymmetries. The gauge field can be made dynamical only for the choice of supersymmetry parameters we made for the vector multiplet. Moreover, we do not have the position dependent coupling factor in front of the hypermultiplet Lagrangian. It can be put in the same form as the vector-multiplet Lagrangian by scaling all the fields in the hypermultiplet by $e^{\frac{\phi}{2}}$. This then introduces a position-dependent mass term for the hypermultiplet scalars. We find it convenient to work with the form of the Lagrangian given in eq. (4.7).

## 4.3 Off-shell interacting hypermultiplet

So far we have only realized supersymmetry on-shell. For a particular choice of supersymmetry parameter we can realize the supersymmetry off-shell. To match eight off-shell fermionic degrees of freedom of hypermultiplet we need four bosonic auxiliary fields $K^{i\bar{j}}$. Supersymmetry can be realized off-shell by appropriately modifying the supersymmetry transformations and the Lagrangian. One adds $K^{\bar{j}i}\nu_j$ to the supersymmetry transformation of $\chi^{\bar{i}}$. Here $\nu^i$ is a spinor of negative chirality which is to be specified in terms of $\xi^i$. This is similar to the use of pure spinors in realizing off-shell supersymmetry for maximally supersymmetric gauge theories. One adds $\tfrac{1}{8}K^{i\bar{j}}K_{i\bar{j}}$ to the Lagrangian. The supersymmetry transformation of $K^{i\bar{j}}$ is then obtained by requiring the modified Lagrangian to be invariant under supersymmetry transformations. This leads to

$$\delta K^{i\bar{j}} = -2\nu^i\slashed{D}\chi^{\bar{j}} - 4\nu^i\psi_j\phi^{j\bar{j}}. \tag{4.13}$$

We relegate the details of the computation of two supersymmetry transformations of fields to appendix B.3. It is not possible to close the supersymmetry algebra off-shell for arbitrary $\nu^i$. We require

$$\nu^i\Gamma^\mu\nu^j + \xi^i\Gamma^\mu\xi^j = 0. \tag{4.14}$$

An explicit example of such $\nu^i$ is given in appendix A.3. Two supersymmetry variations of hypermultiplet fields are computed in appendix B.3. For fermion $\chi^{\bar{i}}$ it take the form of a Lie-derivative and a Weyl transformations as in eq. (3.23) with $\Omega_\chi = \frac{5}{2}$. For scalars and auxiliary fields we get

$$\delta^2 \phi^{i\bar{j}} = \mathcal{L}_\nu \phi^{i\bar{j}} + 2\Omega_\phi \nu^\mu \partial_\mu f \phi^{i\bar{j}} + \nu^i \xi_j K^{j\bar{j}},$$
$$\delta^2 K^{i\bar{j}} = \mathcal{L}_\nu K^{i\bar{j}} + 2\Omega_Y \nu^\mu \partial_\mu f K^{i\bar{j}} - 4\nu^i \xi_j \left( D^2 \phi^{j\bar{j}} - \tfrac{6}{r^2} \phi^{j\bar{j}} + D_k{}^j \phi^{k\bar{j}} + \psi^j \chi^{\bar{j}} \right), \tag{4.15}$$

where $\Omega_\phi = 2$ and $\Omega_Y = 3$. The last term in two variations of the scalar (auxiliary) field is proportional to the EoM for auxiliary (scalar) field. It is straightforward to verify that the Lagrangian is invariant under such a transformation of scalar and auxiliary field. Spinors $\nu^i$, however, can be chosen so that this transformation is trivially zero.

The invariance of the action under $\mathcal{L}_\nu$ follows just as in the case of the vector multiplet. Under a Weyl transformation all but the scalar kinetic and mass terms get scaled by $e^{-6\Omega}$. This cancels against the transformation of $\sqrt{g}$ in the action. The kinetic term for the scalars change as

$$\tfrac{1}{2}\sqrt{g} D_\mu \phi^{i\bar{j}} D^\mu \phi_{i\bar{j}} \to \left( \tfrac{1}{2} D_\mu \phi^{i\bar{j}} D^\mu \phi_{i\bar{j}} + 2\phi^{i\bar{j}} \phi_{i\bar{j}} \partial_\mu \Omega \partial^\mu \Omega - 2\phi^{i\bar{j}} \partial_\mu \phi_{i\bar{j}} \partial^\mu \Omega \right) \tag{4.16}$$

the conformal mass term for the scalar receive extra modification because of the change of the scalar curvature under Weyl transformation

$$\tfrac{3}{r^2} \sqrt{g} \phi^{i\bar{j}} \phi_{i\bar{j}} = \tfrac{R}{10} \sqrt{g} \phi^{i\bar{j}} \phi_{i\bar{j}} \to \sqrt{g} \phi^{i\bar{j}} \phi_{i\bar{j}} \left( \tfrac{R}{10} - 2\partial_\mu \Omega \partial^\mu \Omega - \nabla^2 \Omega \right). \tag{4.17}$$

The second terms above and in the transformation of the scalar kinetic term cancel. The third term combine to give a total derivative and we deduce that the action is invariant.

## 5 Localization of the path integral

In this section we apply the localization procedure for theories constructed in this paper. We will show that for the vector multiplet, the partition function localizes onto solutions of HYM equations, everywhere except at the south pole. Computation of the full partition function needs a detailed knowledge of solutions of HYM equations and is beyond the scope of this paper. Similarly the path integral for the hypermultiplet localizes onto configurations where the gauge-covariant field strengths of the two complex hypermultiplet scalars are related by an almost complex structure on $\mathbb{S}^6$. In the perturbative sector, a simple solution of localization locus is also a solution of EoMs of conformally coupled scalar field.

### 5.1 Vector multiplet

To implement the localization procedure we choose a supercharge $\mathcal{Q}$ which generates supersymmetry transformations w.r.t a specific parameter $\xi^i$ as defined in eq. (3.6). We normalized $\xi^i (\propto \epsilon^i)$ such that

$$\epsilon^{i\dagger} \epsilon^i = 1, \quad \text{no sum over } i. \tag{5.1}$$

We next take the $\mathcal{Q}$-exact Lagrangian to be $\mathcal{Q}V$ where

$$V^{\text{vec}}|_{\text{bos}} \equiv \left( \mathcal{Q}\psi^{i\dagger} \right) \psi^i, \qquad \mathcal{Q}V^{\text{vec}} = \left( \mathcal{Q}\psi^1 \right)^\dagger \left( \mathcal{Q}\psi^1 \right) + \left( \mathcal{Q}\psi^2 \right)^\dagger \left( \mathcal{Q}\psi^2 \right), \tag{5.2}$$

where $\left( \mathcal{Q}\psi^i \right)^\dagger$ is *defined* in terms of the holomorphic fields and the spinor parameter $\xi^i$. We will define it in such a way that it coincides with the complex conjugation of $\mathcal{Q}\psi^i$ along the contour $\{\overline{D^{ij}} = -D_{ij}, \overline{A_\mu} = A_\mu\}$. Along this contour the auxiliary field $D^{12}$ is real while $\overline{D^{11}} = -D^{22}$.

The bosonic part of the standard action of the $(1,0)$ vector multiplet is positive definite along this contour. Terms involving the gauge field are manifestly positive definite. The auxiliary fields contribute $-\frac{1}{4}D^{ij}D_{ij} = \frac{1}{2}\left(\left(D^{12}\right)^2 + |D^{11}|^2\right)$, which is also positive definite.

To proceed we assume

$$\overline{\xi^1} = \xi^2, \quad \overline{\xi^2} = \xi^1. \tag{5.3}$$

Note that this is a consistent condition[8]. Supersymmetry transformations still depend on the two parameters $\xi^1$ and $\xi^2$ holomorphically. We define

$$\left(Q\psi^1\right)^\dagger = \tfrac{1}{2}F_{\mu\nu}\xi^2\Gamma^{\mu\nu} - D^{22}\xi^1 - D^{12}\xi^2, \quad \left(Q\psi^2\right)^\dagger = \tfrac{1}{2}F_{\mu\nu}\xi^1\Gamma^{\mu\nu} + D^{12}\xi^1 + D^{11}\xi^2. \tag{5.4}$$

A straightforward computation then gives[9]

$$
\begin{aligned}
\left(1+\beta^2 x^2\right)\left(Q\psi^1\right)^\dagger\left(Q\psi^1\right) &= \tfrac{1}{2}F_{\mu\nu}F^{\mu\nu} - \tfrac{1}{8}F_{\mu\nu}F_{\rho\sigma}\,\mathcal{I}^{(1)}_{\gamma\delta}\,\varepsilon^{\mu\nu\rho\sigma\gamma\delta} - \tfrac{1}{2}D^{ij}D_{ij} \\
&\quad + \tfrac{1}{2}F_{\mu\nu}\left(D^{11}\epsilon^2\Gamma^{\mu\nu}\epsilon^2 + D^{22}\epsilon^1\Gamma^{\mu\nu}\epsilon^1\right) - D^{12}\left(D^{11}\epsilon^2\epsilon^2 + D^{22}\epsilon^1\epsilon^1\right), \\
\left(1+\beta^2 x^2\right)\left(Q\psi^2\right)^\dagger\left(Q\psi^2\right) &= \tfrac{1}{2}F_{\mu\nu}F^{\mu\nu} - \tfrac{1}{8}F_{\mu\nu}F_{\rho\sigma}\,\mathcal{I}^{(2)}_{\gamma\delta}\,\varepsilon^{\mu\nu\rho\sigma\gamma\delta} - \tfrac{1}{2}D^{ij}D_{ij} \\
&\quad - \tfrac{1}{2}F_{\mu\nu}\left(D^{11}\epsilon^2\Gamma^{\mu\nu}\epsilon^2 + D^{22}\epsilon^1\Gamma^{\mu\nu}\epsilon^1\right) + D^{12}\left(D^{11}\epsilon^2\epsilon^2 + D^{22}\epsilon^1\epsilon^1\right),
\end{aligned}
\tag{5.5}
$$

where we have defined

$$\mathcal{I}^{(i)}_{\mu\nu} \equiv i\epsilon^{\dagger i}\Gamma_{\mu\nu}\epsilon^i. \tag{5.6}$$

$\mathcal{I}^{(i)\,\nu}_{\mu}$ provide two almost complex structures on $\mathbb{S}^6$ which are equal if $\xi^i = i\xi_i$. The vector field $v^\mu$ vanishes identically under this condition. Nevertheless this condition is consistent with eq. (5.3) and $\xi_i = \varepsilon_{ij}\xi^j$. We impose this condition and drop the superscript $^{(i)}$. It is shown in appendix B.4 that $\mathcal{I}_\mu{}^\nu$ satisfies following identities.

$$\mathcal{I}_\mu{}^\nu\mathcal{I}_\nu{}^\rho = -\delta_\mu{}^\rho, \qquad \varepsilon^{\mu\nu\rho\sigma\gamma\delta}\mathcal{I}_{\rho\sigma}\mathcal{I}_{\gamma\delta} = -8\mathcal{I}^{\mu\nu}. \tag{5.7}$$

Combining the two terms in eq. (5.5) we get the following simple form for the localization Lagrangian.

$$QV^{\text{vec}} = \frac{1}{1+\beta^2 x^2}\left(F_{\mu\nu}F^{\mu\nu} - \tfrac{1}{4}F_{\mu\nu}F_{\rho\sigma}\,\mathcal{I}_{\gamma\delta}\,\varepsilon^{\mu\nu\rho\sigma\gamma\delta} - D^{ij}D_{ij}\right), \tag{5.8}$$

We next write the localization Lagrangian in a manifestly positive-definite form. The term involving auxiliary fields is already positive-definite as argued earlier. For the term involving the vector field, we decompose the 2-form field strength w.r.t to the almost complex structure as follows.

$$F_{\mu\nu} = F^+_{\mu\nu} + F^-_{\mu\nu} + \tfrac{1}{6}F^0\mathcal{I}_{\mu\nu}, \qquad F^\pm = \pm \star\left(F \wedge \mathcal{I}\right). \tag{5.9}$$

where $F^{+(-)}$ has six(eight) independent components. More explicitly

$$
\begin{aligned}
F^+_{\mu\nu} &= \tfrac{1}{2}\left(F_{\mu\nu} + \mathcal{I}_\mu{}^\rho\mathcal{I}_\nu{}^\sigma F_{\rho\sigma} - \tfrac{1}{3}\mathcal{I}_{\mu\nu}\left(\mathcal{I}^{\rho\sigma}F_{\rho\sigma}\right)\right), \\
F^-_{\mu\nu} &= \tfrac{1}{2}\left(F_{\mu\nu} - \mathcal{I}_\mu{}^\rho\mathcal{I}_\nu{}^\sigma F_{\rho\sigma}\right), \\
F^0 &= \mathcal{I}^{\rho\sigma}F_{\rho\sigma}.
\end{aligned}
\tag{5.10}
$$

---

[8]For $\mathcal{N} = 1$ theory on $\mathbb{S}^4$ such a condition will not be consistent because $\xi^1$ and $\xi^2$ have opposite chirality. This remains a key obstacle in doing a localization analysis for $\mathcal{N} = 1$ supersymmetry on $\mathbb{S}^4$.

[9]Bilinear in this equation are simply inner products of spinors and various Gamma matrices, without the usual insertion of the charge conjugation matrix.

The localization Lagrangian can now be written as

$$\mathcal{Q}V^{\text{vec}} = \frac{1}{1 + \beta^2 x^2}\left(2\left(F^-\right)^2 + \tfrac{1}{2}\left(F^0\right)^2 + 2\left(|D^{12}|^2 + |D^{11}|^2\right)\right). \tag{5.11}$$

Hence the localization locus is given by

$$F^0 = 0, \quad F^- = 0, \quad D^{ij} = 0, \qquad \text{everywhere expect at the south pole.} \tag{5.12}$$

First two equations are called Hermitian Yang-Mills equations [29]. These are natural analogues of (anti)-self-duality condition in four dimensions which describe instanton configurations. For the case of six-sphere these are studied in [30, 31]. We leave a detailed analysis of the localization locus and computation of partition function for future work.

## 5.2 Hypermultiplet

Localization analysis for the hypermultiplet can also be performed in a completely analogous manner. We choose a localization Lagrangian $\mathcal{Q}V^{\text{hyp}} \propto \left(\mathcal{Q}\chi^i\right)^\dagger \mathcal{Q}\chi^i$, with $\mathcal{Q}$ being the same supercharge as the one used in the case of vector multiplet. $\left(\mathcal{Q}\chi^i\right)^\dagger$ is defined as

$$\left(\mathcal{Q}\chi^i\right)^\dagger = 2\mathcal{O}_\mu\phi^2_{\bar{i}}\xi^1\Gamma^\mu + 2\mathcal{O}_\mu\phi^1_{\bar{i}}\xi^2\Gamma^\mu + K^2_{\bar{i}}\nu^1 + K^1_{\bar{i}}\nu^2, \tag{5.13}$$

where $\mathcal{O}_\mu = D_\mu + 4\partial_\mu f$. This definition coincides with the complex conjugation along the contour $\{\overline{\phi^{ij}} = \phi_{i\bar{j}}, \overline{K^{ij}} = K_{ij}\}$. We choose the spinors $\nu^i$ as $\nu^1 = C\xi^2, \quad \nu^2 = C\xi^1$, where $C$ is the charge conjugation matrix. A straightforward computation gives the following

$$\left(1 + \beta^2 x^2\right)\mathcal{Q}V^{\text{hyp}} = 4\mathcal{O}_\mu\phi^{i\bar{j}}\mathcal{O}^\mu\phi_{i\bar{j}} - 8\mathcal{I}^{\mu\nu}\left(\mathcal{O}_\mu\phi^{1\bar{2}}\mathcal{O}_\nu\phi^{1\bar{1}} + \mathcal{O}_\mu\phi_{1\bar{2}}O_\nu\phi_{1\bar{1}}\right) + K^{i\bar{j}}K_{i\bar{j}}. \tag{5.14}$$

The term involving auxiliary fields is positive definite. First two terms can also be written as a manifestly positive definite form by completing the square

$$\left(1 + \beta^2 x^2\right)\mathcal{Q}V^{\text{hyp}} = 8\left(\mathcal{O}_\mu\phi^{1\bar{2}} - \mathcal{I}_\mu{}^\nu\mathcal{O}_\nu\phi_{1\bar{1}}\right)\left(\mathcal{O}^\mu\phi_{1\bar{2}} - \mathcal{I}^{\mu\nu}\mathcal{O}_\nu\phi^{1\bar{1}}\right) + K^{i\bar{j}}K_{i\bar{j}}. \tag{5.15}$$

Auxiliary fields vanish at the localization locus and scalar fields satisfy

$$\mathcal{O}_\mu\phi^{1\bar{2}} - \mathcal{I}_\mu{}^\nu\mathcal{O}_\nu\phi_{1\bar{1}} = 0, \quad \text{everywhere except the south pole.} \tag{5.16}$$

This is one complex and two real constraints. We do not attempt to solve for a general solution to this constraint here. A simple solution is $\mathcal{O}_\mu\phi^{ij} = 0$. This corresponds to

$$\phi_{ij} = C_{ij}\beta\left(1 + \beta^2 x^2\right)^2, \tag{5.17}$$

where $C_{ij}$ are dimensionless constants valued in the appropriate representation of the Lie algebra. This $\phi_{ij}$ is actually a solution of the EoMs of the classical action. The value of the classical action, however, is divergent unless $C_{ij}$ vanish identically. Moreover one can explicitly check that this a UV divergence localized at the south pole where the value of the scalar field diverges. So the path integral for the hypermultiplet does not receive contribution from this simple solution of eq. (5.16).

We conclude this section by showing that for the fields satisfying eq. (5.16) the supersymmetry variation of hypermultiplet fermions is zero. We focus on $\delta\chi^1$, computation for $\delta\chi^2$ is analogous. We argue this point by showing that all bilinears of $\delta\chi^1$ with $\xi^1$ and $\xi^{1c} \equiv C\overline{\xi^1}$

vanish. We first show this for $\xi^{1c}$. The even ranked bilinear vanish trivially as both $\delta\chi^{\bar{1}}$ and $\xi^{1c}$ have negative chirality. A short computation gives

$$\xi^{1c}\Gamma^{\mu}\delta\chi^{\bar{1}} = -\left(\mathcal{O}^{\mu}\phi^{2\bar{1}} + \mathcal{I}^{\mu\nu}\mathcal{O}_{\nu}\phi^{1\bar{1}}\right) - i\left(\mathcal{O}^{\mu}\phi^{1\bar{1}} - \mathcal{I}^{\mu\nu}\mathcal{O}_{\nu}\phi^{2\bar{1}}\right), \tag{5.18}$$

which vanishes on the locus eq. (5.16).

For the rank-3 bilinear we get

$$\xi^{1c}\Gamma^{\mu\nu\rho}\delta\chi^{\bar{1}} = -\left(\mathcal{O}_{\sigma}\phi^{2\bar{1}} - i\mathcal{O}_{\sigma}\phi^{1\bar{1}}\right)\left(\tfrac{1}{2}\varepsilon^{\mu\nu\rho\sigma\alpha\beta}\mathcal{I}_{\alpha\beta} - i\left(\mathcal{I}^{\nu\rho}\delta^{\mu\sigma} + \text{cyc. perms.}\right)\right). \tag{5.19}$$

Using the second identity in eq. (5.7) we can write

$$\tfrac{1}{2}\varepsilon^{\mu\nu\rho\sigma\alpha\beta}\mathcal{I}_{\alpha\beta} = -\mathcal{I}^{\mu\nu}\mathcal{I}^{\rho\sigma} + \text{cyc. perms. of } \{\mu,\nu,\rho\}, \tag{5.20}$$

Using this we get

$$\xi^{1c}\Gamma^{\mu\nu\rho}\delta\chi^{\bar{1}} = -\mathcal{I}^{\nu\rho}\left(\mathcal{O}^{\mu}\phi^{1\bar{1}} - \mathcal{I}^{\mu\sigma}\mathcal{O}_{\sigma}\phi^{2\bar{1}}\right) - i\mathcal{I}^{\nu\rho}\left(\mathcal{O}^{\mu}\phi^{1\bar{2}} + \mathcal{I}^{\mu\sigma}\mathcal{O}_{\sigma}\phi^{1\bar{1}}\right) + \text{cyc. perms.}, \tag{5.21}$$

which again vanishes on the locus in eq. (5.16).

For $\xi^1$, the odd-ranked bilinears are identically zero due to the opposite chirality. The zero-ranked bilinear is proportional to the vector field $v^{\mu}$ and hence vanishes identically without using the localization locus equations. For the rank-2 bilinear we use Fierz identities to reduce it to the bilinear that we have already computed.

$$\begin{aligned}
\xi^1\Gamma^{\mu\nu}\delta\chi^{\bar{1}} = \xi^1\Gamma^{\mu}\Gamma^{\nu}\delta\chi^{\bar{1}} &\propto \xi^1\Gamma^{\mu}\Gamma^{\nu}\delta\chi^{\bar{1}}\left(\xi^{1c}\xi^1\right) \\
&= \tfrac{1}{4}\left(\xi^1\Gamma^{\mu}\xi^1\right)\left(\xi^{1c}\Gamma^{\nu}\delta\chi^{\bar{1}}\right) - \tfrac{1}{8}\left(\xi^1\Gamma^{\alpha\beta\mu}\xi^1\right)\left(\xi^{1c}\Gamma^{\alpha\beta\nu}\delta\chi^{\bar{1}}\right) \\
&\quad + \tfrac{1}{4}\left(\xi^1\Gamma^{\alpha\mu\nu}\xi^1\right)\left(\xi^{1c}\Gamma_{\alpha}\delta\chi^{\bar{1}}\right) = 0.
\end{aligned} \tag{5.22}$$

In the first line we have proportional sign because $\left(\xi^{1c}\xi^1\right) = \frac{1}{1+\beta^2 x^2}$. The second line follows from the Fierz identity on the third line of eq. (A.13). All bilinears on the second line involve $\xi^{1c}$ and $\delta\chi^{\bar{1}}$ and vanish as we have shown earlier.

# 6 Conclusions and outlook

In this paper we constructed theories on $\mathbb{S}^6$ with (1,0) supersymmetry. We wrote down explicit Lagrangians for vector and hypermultiplets with off-shell supersymmetry. We also determined the localization locus for the path integral in these theories. We showed that the path integral for the vector multiplet localizes onto solutions of Hermitian Yang-Mills equations. In the perturbative sector we showed that the path integral for the hypermultiplet localizes to field configurations where the field strength of two hypermultiplet scalars is related to each other. There are a number of issues left open for future research on which we comment now.

First one would like to compute the partition function and other supersymmetric observables for these theories. This requires a complete analysis of the localization loci derived above. One also needs to understand how to take into account the configurations localized at the south pole. It will be worthwhile to explore various approaches to this problem. One can modify the Lagrangian near the south pole to capture the contribution from the singular scalar field configurations discussed in previous section. Such singularities are generally related to operator insertions at the south pole. This is similar to the singular field configurations produced by surface operators in $\mathcal{N} = 4$ SYM [32]. It is also possible to use a different supercharge for localization which may give a different and simpler localization locus. This may also be

achieved by suitably modifying the localization term. Computation of partition function in the non-perturbative sector is even more challenging. Non-perturbative configurations of gauge fields in higher dimensions are not well understood[10]. Any progress in this direction will be an important step towards understanding the structure of partition function of these theories.

A possible extension of this work is to construct supersymmetric theories with non-constant coupling on spheres of different dimensions. A simple example will be 6D $(1, 1)$ theory which arises by taking the hypermultiplet in our construction to be in the adjoint representation of the gauge group. This would be a construction of the maximally supersymmetric theory on $\mathbb{S}^6$ different than the one given in [7]. This hints at the existence of different formulations of maximally supersymmetric theories in other dimensions. One can consider the dimensional reduction of 10D SYM but allowing for a non-constant coupling. We report this analysis in a companion paper [33].

Another interesting issue is to extend our analysis to include tensor multiplets on $\mathbb{S}^6$. For consistent theories in 6D with different gauge groups one needs to include a certain number of tensor multiplets [34, 35]. Due to the self-duality constraint on the 3-form field strength the Lagrangian for the tensor multiplet cannot be written in a simple manner. However it is reasonable to expect that the analysis of supersymmetry can be performed at the levels of EoMs and the correct form of supersymmetry transformations can be determined. This would be of significant importance because the detailed structure of the partition function depends on the supersymmetry transformations (à la localization) and the actual Lagrangian only appears in the weighting factor multiplying contributions to the path integral from different loci. We hope to explore these issues in future.

# 7 Acknowledgements

I am thankful to Joseph Minahan for numerous helpful discussions over the course of this project and comments on an earlier draft. This material is based upon work supported by the U.S. Department of Energy, Office of Science, Office of High Energy Physics of U.S. Department of Energy under grant Contract Number DE-SC0012567 and the fellowship by the Knut and Alice Wallenberg Foundation, Stockholm Sweden.

# A  Clifford algebra conventions

## A.1  Clifford algebra in even dimensions

Here we list our spinor conventions and useful properties of Clifford algebra in 4D and 6D Euclidean space. We start with some generalities about Clifford algebra $d = 2n$ dimensions.

The charge conjugation matrix $C$ and gamma matrices satisfy the following:

$$C^{\mathrm{tr}} = (-)^{\frac{n(n+1)}{2}} C, \quad C^* = (-)^{\frac{n(n+1)}{2}} C^{-1}, \qquad \Gamma_\mu^{\mathrm{tr}} = (-)^n C^{-1}\Gamma_\mu C = \Gamma_\mu^*. \tag{A.1}$$

From these one can derive following useful identities

$$\left(\Gamma_{\mu_1\mu_2\cdots\mu_k}\right)^{\mathrm{tr}} = (-)^{nk}(-)^{\frac{k(k-1)}{2}} C^{-1}\Gamma_{\mu_1\mu_2\cdots\mu_k} C = (-)^{\frac{k(k-1)}{2}} \Gamma^*_{\mu_1\mu_2\cdots\mu_k} = (-)^{nk} C^{-1}\left(\Gamma_{\mu_1\mu_2\cdots\mu_k}\right)^\dagger C. \tag{A.2}$$

Covariantly transforming bilinears of spinors are defined as follow:

$$(\psi\chi) \equiv \psi^{\mathrm{tr}}C^{-1}\chi. \tag{A.3}$$

---

[10]We thank D. Harland for pointing out that there is precisely one known solution of HYM equations on $\mathbb{S}^6$ given in [30].

Using this definition and anti-commutation relations of gamma matrices we can obtain the following identity:

$$\psi\Gamma^{\mu_1}\Gamma^{\mu_2}\cdots\Gamma^{\mu_k}\chi \;=\; \sigma\,(-)^{nk}(-)^{\frac{n(n+1)}{2}}\,\chi\Gamma^{\mu_k}\Gamma^{\mu_{k-1}}\cdots\Gamma^{\mu_1}\psi, \tag{A.4}$$

where $\sigma = +(-)$ for commuting (anti-commuting) spinors. From this identity one can immediately derive the following.

$$\psi\Gamma^{\mu_1\mu_2\cdots\mu_k}\chi \;=\; \sigma\,(-)^{\frac{k(k-1)}{2}}(-)^{nk}(-)^{\frac{n(n+1)}{2}}\,\chi\Gamma^{\mu_1\mu_2\cdots\mu_k}\psi. \tag{A.5}$$

For 4D and 6D, following special cases of above identities will be frequently used.

$$\psi\Gamma^{\mu}\chi=-\sigma\chi\Gamma^{\mu}\psi, \quad \psi\Gamma^{\mu_1\mu_2\mu_3}\chi = +\sigma\chi\Gamma^{\mu_1\mu_2\mu_3}\psi, \quad \psi\Gamma^{\mu_1}\Gamma^{\mu_2}\Gamma^{\mu_3}\chi = -\sigma\chi\Gamma^{\mu_3}\Gamma^{\mu_2}\Gamma^{\mu_1}\psi. \tag{A.6}$$

One can define a chirality matrix $\Gamma$ in even dimensions as follows:

$$\Gamma \equiv (-i)^n\,\Gamma^{12\cdots d}. \tag{A.7}$$

For even $n$, the irreducible chiral representation of Spin(2n) is real and for odd $n$ it is complex. A useful identity relating different clifford algebra elements is

$$\Gamma_{\mu_1\mu_2\cdots\mu_p} = \frac{i^n}{(2n-p)!}\varepsilon_{\mu_1\mu_2\cdots\mu_d}\Gamma\Gamma^{\mu_d\cdots\mu_p+1}\,. \tag{A.8}$$

For two positive chirality spinors, we have the following identity:

$$\psi_\pm\Gamma^{\mu_1}\Gamma^{\mu_1}\cdots\Gamma^{\mu_k}\chi_\pm \;=\; (-)^{n+k}\,\psi_\pm\Gamma^{\mu_1}\Gamma^{\mu_1}\cdots\Gamma^{\mu_k}\chi_\pm, \tag{A.9}$$

where the subscript denote the chirality of the spinor. For opposite chirality spinors we have an extra minus sign on the right hand sign.

## A.2 Fierz Identities

For 4D we have the basic Fierz identity which can be derived by usual methods:

$$\chi\,\psi^{\mathrm{tr}} = \tfrac{1}{4}(\psi\chi)\,C + \tfrac{1}{4}(\psi\Gamma^\mu\chi)\Gamma^\mu C - \tfrac{1}{8}(\psi\Gamma^{\mu\nu}\chi)\Gamma_{\mu\nu}C - \tfrac{1}{4}(\psi\Gamma^\mu\Gamma\chi)\Gamma^\mu\Gamma C + \tfrac{1}{4}(\psi\Gamma\chi)\Gamma C. \tag{A.10}$$

From this we can obtain following four identities:

$$\begin{aligned}
\chi_\pm(\psi_\pm\eta_\mp) &= 0 = -\tfrac{1}{8}(\psi_\pm\Gamma^{\mu\nu}\chi_\pm)\Gamma_{\mu\nu}\eta_\mp,\\
\chi_\pm(\psi_\pm\eta_\pm) &= \tfrac{1}{2}(\psi_\pm\chi_\pm)\eta_\pm + \tfrac{1}{8}(\psi_\pm\Gamma^{\mu\nu}\chi_\pm)\Gamma_{\mu\nu}\eta_\pm,\\
\chi_\mp(\psi_\pm\eta_\pm) &= \tfrac{1}{2}(\psi_\pm\Gamma^\mu\chi_\mp)\Gamma_\mu\eta_\pm.
\end{aligned} \tag{A.11}$$

In 6D the basic Fierz identity takes the following form :

$$\begin{aligned}
\chi\psi^{\mathrm{tr}} = {}& \tfrac{1}{8}(\psi\chi)\,C + \tfrac{1}{8}(\psi\Gamma^\mu\chi)\Gamma_\mu C - \tfrac{1}{16}(\psi\Gamma^{\mu\nu}\chi)\Gamma_{\mu\nu}C - \tfrac{1}{6\times8}(\psi\Gamma^{\mu\nu\rho}\chi)(\Gamma_{\mu\nu\rho}C)\\
&- \tfrac{1}{8}(\chi\Gamma\psi)(\Gamma C) - \tfrac{1}{8}(\psi\Gamma^\mu\Gamma\chi)(\Gamma_\mu\Gamma C) + \tfrac{1}{16}(\psi\Gamma^{\mu\nu}\Gamma\chi)(\Gamma_{\mu\nu}\Gamma C)\,.
\end{aligned} \tag{A.12}$$

From this we obtain following identities for 6D spinors

$$\begin{aligned}
\chi_\pm(\psi_\pm\eta_\pm) &= 0 = -\tfrac{1}{48}(\psi_\pm\Gamma^{\mu\nu\rho}\chi_\pm)\Gamma_{\mu\nu\rho}\eta_\pm,\\
\chi_\pm(\psi_\pm\eta_\mp) &= \tfrac{1}{4}(\psi_\pm\Gamma^\mu\chi_\pm)\Gamma_\mu\eta_\mp - \tfrac{1}{48}(\psi_\pm\Gamma^{\mu\nu\rho}\chi_\pm)\Gamma_{\mu\nu\rho}\eta_\mp,\\
\chi_\mp(\psi_\pm\eta_\mp) &= \tfrac{1}{4}(\psi_\pm\chi_\mp)\eta_\mp - \tfrac{1}{8}(\psi_\pm\Gamma^{\mu\nu}\chi_\mp)\Gamma_{\mu\nu}\eta_\mp.
\end{aligned} \tag{A.13}$$

### A.3 Explicit gamma matrices and Killing Spinors

A set of explicit Gamma matrices in Euclidean 6D is given by

$$
\Gamma^1 = \begin{pmatrix} 1 & 0 & 0 & 0 & 0 & 0 & 0 & 0 \\ 0 & -1 & 0 & 0 & 0 & 0 & 0 & 0 \\ 0 & 0 & -1 & 0 & 0 & 0 & 0 & 0 \\ 0 & 0 & 0 & 1 & 0 & 0 & 0 & 0 \\ 0 & 0 & 0 & 0 & -1 & 0 & 0 & 0 \\ 0 & 0 & 0 & 0 & 0 & 1 & 0 & 0 \\ 0 & 0 & 0 & 0 & 0 & 0 & 1 & 0 \\ 0 & 0 & 0 & 0 & 0 & 0 & 0 & -1 \end{pmatrix} \quad
\Gamma^2 = \begin{pmatrix} 0 & -i & 0 & 0 & 0 & 0 & 0 & 0 \\ i & 0 & 0 & 0 & 0 & 0 & 0 & 0 \\ 0 & 0 & 0 & i & 0 & 0 & 0 & 0 \\ 0 & 0 & -i & 0 & 0 & 0 & 0 & 0 \\ 0 & 0 & 0 & 0 & 0 & i & 0 & 0 \\ 0 & 0 & 0 & 0 & -i & 0 & 0 & 0 \\ 0 & 0 & 0 & 0 & 0 & 0 & 0 & -i \\ 0 & 0 & 0 & 0 & 0 & 0 & i & 0 \end{pmatrix}
$$

$$
\Gamma^3 = \begin{pmatrix} 0 & 0 & 1 & 0 & 0 & 0 & 0 & 0 \\ 0 & 0 & 0 & 1 & 0 & 0 & 0 & 0 \\ 1 & 0 & 0 & 0 & 0 & 0 & 0 & 0 \\ 0 & 1 & 0 & 0 & 0 & 0 & 0 & 0 \\ 0 & 0 & 0 & 0 & 0 & 0 & -1 & 0 \\ 0 & 0 & 0 & 0 & 0 & 0 & 0 & -1 \\ 0 & 0 & 0 & -1 & 0 & 0 & 0 & 0 \\ 0 & 0 & 0 & 0 & 0 & -1 & 0 & 0 \end{pmatrix} \quad
\Gamma^4 = \begin{pmatrix} 0 & 0 & -i & 0 & 0 & 0 & 0 & 0 \\ 0 & 0 & 0 & -i & 0 & 0 & 0 & 0 \\ i & 0 & 0 & 0 & 0 & 0 & 0 & 0 \\ 0 & i & 0 & 0 & 0 & 0 & 0 & 0 \\ 0 & 0 & 0 & 0 & 0 & 0 & i & 0 \\ 0 & 0 & 0 & 0 & 0 & 0 & 0 & i \\ 0 & 0 & 0 & 0 & -i & 0 & 0 & 0 \\ 0 & 0 & 0 & 0 & 0 & -i & 0 & 0 \end{pmatrix}. \tag{A.14}
$$

$$
\Gamma^5 = \begin{pmatrix} 0 & 0 & 0 & 0 & 1 & 0 & 0 & 0 \\ 0 & 0 & 0 & 0 & 0 & 1 & 0 & 0 \\ 0 & 0 & 0 & 0 & 0 & 0 & 1 & 0 \\ 0 & 0 & 0 & 0 & 0 & 0 & 0 & 1 \\ 1 & 0 & 0 & 0 & 0 & 0 & 0 & 0 \\ 0 & 1 & 0 & 0 & 0 & 0 & 0 & 0 \\ 0 & 0 & 1 & 0 & 0 & 0 & 0 & 0 \\ 0 & 0 & 0 & 1 & 0 & 0 & 0 & 0 \end{pmatrix} \quad
\Gamma^6 = \begin{pmatrix} 0 & 0 & 0 & 0 & -i & 0 & 0 & 0 \\ 0 & 0 & 0 & 0 & 0 & -i & 0 & 0 \\ 0 & 0 & 0 & 0 & 0 & 0 & -i & 0 \\ 0 & 0 & 0 & 0 & 0 & 0 & 0 & -i \\ i & 0 & 0 & 0 & 0 & 0 & 0 & 0 \\ 0 & i & 0 & 0 & 0 & 0 & 0 & 0 \\ 0 & 0 & i & 0 & 0 & 0 & 0 & 0 \\ 0 & 0 & 0 & i & 0 & 0 & 0 & 0 \end{pmatrix}
$$

The indices on gamma matrices are the *flat* ones. A spinor of positive(negative) chirality has the general form

$$
\epsilon_\pm = (a, \mp a, b, \pm b, c, \pm c, d, \mp d). \tag{A.15}
$$

For the hypermultiplet a supersymmetry is realized off-shell if we choose $\epsilon^1$ and $\nu^1$ to be the positive and negative chirality spinors specified by

$$
a = e^{i\theta_1}\cos(\Theta), \quad b = e^{i\theta_2}\sin(\Theta)\cos(\Psi),
$$
$$
c = e^{i\theta_2}\sin(\Theta)\cos(X)\sin(\Psi), \quad d = e^{i\theta_1}\sin(\Theta)\sin(X)\sin(\Psi). \tag{A.16}
$$

We choose $\epsilon^2$ and $\nu^2$ to be complex conjugates of $\epsilon^1$ and $\nu^1$ respectively.

To carry out localization analysis we imposed another consistence condition on supersymmetry parameters, i.e., $\xi^i = i\xi_i$. Spinor parameters which satisfy this requirement are specified by.

$$
(\theta_1, \theta_2) = \{\left(\tfrac{\pi}{4}, \tfrac{\pi}{4}\right), \left(\tfrac{\pi}{4}, -\tfrac{3\pi}{4}\right), \left(-\tfrac{3\pi}{4}, \tfrac{\pi}{4}\right), \left(-\tfrac{3\pi}{4}, -\tfrac{3\pi}{4}\right)\}. \tag{A.17}
$$

## B Technical details of some computations

In this appendix we give details of some computations whose results are used in the paper.

### B.1 Covariantized supersymmetry variation of covariantized Lagrangian in 4D and 6D

Covariantized Lagrangian in both 4D and 6D has the following form:

$$
\mathcal{L} = \tfrac{1}{4}F^2 + \tfrac{1}{2}\psi^i \slashed{\nabla}\psi_i - \tfrac{1}{4}D^{ij}D_{ij}. \tag{B.1}
$$

In 4D, $\psi^1$ has positive chirality while $\psi^2$ have negative chirality. In 6D, however, they both have positive chirality. The supersymmetry variation of the first and the last term in the Lagrangian is straightforward to evaluate:

$$
\delta\left(\tfrac{1}{4}F^2\right) = F^{\mu\nu}\nabla_\mu\left(\xi^i\Gamma_\nu\psi_i\right),
$$
$$
\delta\left(-\tfrac{1}{4}D^{ij}D_{ij}\right) = -D^{ij}\left(\xi_i\slashed{\nabla}\psi_j\right). \tag{B.2}
$$

The supersymmetry variation of the kinetic term for fermion takes the following form:

$$\delta \left( \tfrac{1}{2} \psi^i \slashed{\nabla} \psi_i \right) = \delta \psi^i \slashed{\nabla} \psi_i + \tfrac{1}{2} \nabla_\mu \left( \delta \psi_i \Gamma^\mu \psi^i \right), \tag{B.3}$$

where we have kept a total derivative term for future reference. After using $\delta \psi^i$, the first term becomes

$$\delta \psi^i \slashed{\nabla} \psi_i = -F_{\mu\nu} \xi^i \Gamma^\nu \nabla^\mu \psi_i + D^{ij} \left( \xi_i \slashed{\nabla} \psi_j \right) - \tfrac{1}{2} \xi^i \Gamma^{\nu\mu\rho} \nabla_\rho \psi_i F_{\mu\nu}. \tag{B.4}$$

Using Bianchi identity for the gauge field, the last term on the r.h.s can be written as

$$-\tfrac{1}{2} \xi^i \Gamma^{\nu\mu\rho} \nabla_\rho \psi_i F_{\mu\nu} = \tfrac{1}{2} F_{\mu\nu} \nabla_\rho \xi^i \Gamma^{\nu\mu\rho} \psi_i - \tfrac{1}{2} \nabla_\rho \left( \xi^i \Gamma^{\nu\mu\rho} \psi_i F_{\mu\nu} \right). \tag{B.5}$$

Combining all these terms, the total change in the Lagrangian becomes

$$\begin{aligned} \delta \mathcal{L} &= F^{\mu\nu} \nabla_\mu \xi^i \Gamma_\nu \psi_i + \tfrac{1}{2} F_{\mu\nu} \nabla_\rho \xi^i \Gamma^{\nu\mu\rho} \psi_i + \text{t.d} = \tfrac{1}{2} F^{\mu\nu} \left( \nabla_\rho \xi^i \Gamma^{\nu\mu} \Gamma^\rho \psi_i \right) + \text{t.d} \\ &= \tfrac{1}{2} F_{\mu\nu} \left( \psi_i \Gamma^\rho \Gamma^{\mu\nu} \nabla_\rho \psi^i \right) + \text{t.d.} \end{aligned} \tag{B.6}$$

The last equality follows by using the identity in eq. (A.6) for anti-commuting spinors. t.d denotes the total derivative term

$$\text{t.d} = \tfrac{1}{2} \nabla_\mu \left( \delta \psi_i \Gamma^\mu \psi^i \right) - \tfrac{1}{2} \nabla_\rho \left( \xi^i \Gamma^{\nu\mu\rho} \psi_i F_{\mu\nu} \right). \tag{B.7}$$

Using $\delta \psi_i$, the first term on the r.h. s becomes:

$$\tfrac{1}{2} \nabla_\mu \left( \delta \psi_i \Gamma^\mu \psi^i \right) = -\tfrac{1}{4} \nabla_\mu \left( \xi^i \Gamma^{\mu\rho\,\nu} \psi_i \right) F_{\rho\sigma} + \tfrac{1}{2} \nabla_\mu \left( F^{\mu\nu} \xi^i \Gamma_\nu \psi_i \right) - \tfrac{1}{2} \nabla_\mu \left( D^{ij} \xi_i \Gamma^\mu \psi_j \right), \tag{B.8}$$

so the total derivative term is

$$\text{t.d} = -\tfrac{1}{4} \nabla_\rho \left( F_{\mu\nu} \left( \psi_i \Gamma^{\rho\mu\nu} \xi^i \right) \right) + \tfrac{1}{2} \nabla_\mu \left( F^{\mu\nu} \left( \psi_i \Gamma_\nu \xi^i \right) \right) - \tfrac{1}{2} \nabla_\rho \left( D^{ij} \left( \xi_i \Gamma^\rho \psi_j \right) \right). \tag{B.9}$$

Note that the results in eqs. (B.6) and (B.9) hold in 4D and 6D.

## B.2 Two covariantized supersymmetry variations in 4D and 6D

Now we compute the change in the fields under two successive covariantized supersymmetry transformations. We will find the following two relations useful in our computations:

$$\xi^i \Gamma^\mu \xi^j = \tfrac{1}{2} v^\mu \varepsilon^{ij}, \qquad \xi^i \Gamma^{\mu\nu\rho} \xi^j = \xi^j \Gamma^{\mu\nu\rho} \xi^i. \tag{B.10}$$

It follows from the second relation that $\xi^i \Gamma^{\mu\nu\rho} \xi_i = 0$.

### (1) The gauge field

Using the supersymmetry transformation of gauge field and fermion given in eq. (2.6) one gets

$$\delta^2 A_\mu = -\tfrac{1}{2} F^{\nu\rho} \xi^i \Gamma_\mu \Gamma^{\nu\rho} \xi_i - D^{ij} \xi^i \Gamma_\mu \xi_j. \tag{B.11}$$

Using eq. (B.10), this becomes

$$\delta^2 A_\mu = -v^\nu F_{\mu\nu} = \mathcal{L}_v A - \partial_\mu \left( A_\nu v^\nu \right). \tag{B.12}$$

Since we did not assume any specific form of the supersymmetry parameter, the above relation holds in general, as long as the supersymmetry transformations of gauge field and the fermion have the form given in eq. (2.6).

### (2) The gaugino

Using the covriantized supersymmetry transformations of eq. (2.6), we get the following:

$$
\begin{aligned}
\delta^2 \psi^i &= \Gamma^{\mu\nu}\xi^i\left(\psi_j\Gamma_\nu\nabla_\mu\xi^j\right)+\xi^i\left(\xi^j\slashed{\nabla}\psi_j\right)-\Gamma^\mu\Gamma^\nu\xi^i\left(\xi^j\Gamma_\nu\nabla_\mu\psi_j\right) \\
&\quad +\xi_j\left(\xi^i\slashed{\nabla}\psi^j\right)+\xi_j\left(\xi^j\slashed{\nabla}\psi^i\right).
\end{aligned}
\tag{B.13}
$$

For simplicity, let us do the computation for $i=1$. The $i=2$ case is completely analogous. We get:

$$
\begin{aligned}
\delta^2 \psi^1 &= \Gamma^{\mu\nu}\xi^1\left(\psi_j\Gamma_\nu\nabla_\mu\xi^j\right)-2\xi^1\left(\xi^1\slashed{\nabla}\psi^2\right)+\Gamma^\nu\Gamma^\mu\xi^1\left(\xi^1\Gamma_\nu\nabla_\mu\psi^2\right) \\
&\quad +2\xi^2\left(\xi^1\slashed{\nabla}\psi^1\right)-\Gamma^\nu\Gamma^\mu\xi^1\left(\xi^2\Gamma_\nu\nabla_\mu\psi^1\right).
\end{aligned}
\tag{B.14}
$$

Let us now specialize to 4D and 6D separately to simplify the above expression.

### 4D

In this case, the first term on the second line in eq. (B.14) is zero and we can write:

$$
\delta^2 \psi^1 = \Gamma^{\mu\nu}\xi^1\left(\psi_j\Gamma_\nu\nabla_\mu\xi^j\right)-\Gamma^\mu\Gamma^\nu\xi^1\left(\xi^1\Gamma_\nu\nabla_\mu\psi^2\right)-\Gamma^\nu\Gamma^\mu\xi^1\left(\xi^2\Gamma_\nu\nabla_\mu\psi^1\right).
\tag{B.15}
$$

Now we use Fierz identities of eq. (A.11) to see that

$$
\begin{aligned}
\Gamma^\mu\Gamma^\nu\xi^1\left(\xi^1\Gamma_\nu\nabla_\mu\psi^2\right) &= 2\Gamma^\mu\nabla_\mu\psi^2\left(\xi^1\xi^1\right)=0, \\
\Gamma^\nu\Gamma^\mu\xi^1\left(\xi^2\Gamma_\nu\nabla_\mu\psi^1\right) &= 2\nabla_\mu\psi^1\left(\xi^1\Gamma^\mu\xi^2\right).
\end{aligned}
\tag{B.16}
$$

The second equality on the first line follows from eq. (A.6) because $\xi^1$ is a commuting spinor. So

$$
\delta^2 \psi^1 = \nu^\mu\nabla_\mu\psi^1+\Gamma^{\mu\nu}\xi^1\left(\psi_j\Gamma_\nu\nabla_\mu\xi^j\right).
\tag{B.17}
$$

For $\psi^2$ we get a similar expression with $1\to 2$ in the above equation.

If the supersymmetry parameters are Killing spinors then the second term on the r.h.s of $\delta^2\psi^1$ vanishes. Moreover one can show that $\nu^\mu$ is a Killing vector in this case. Hence $\delta^2\psi^i$ is equal to the Lie derivative of the spinor $\psi^i$ along a Killing vector.

If $\xi^i$ satisfy CKS equation, $\nabla_\mu\xi^i=\Gamma_\mu\tilde{\xi}^i=\beta\Gamma_\mu\xi_i$, then we need to analyze the second term appearing on the r.h.s of $\delta^2\psi^1$ further. This term can be written as

$$
\begin{aligned}
\Gamma^{\mu\nu}\xi^1\left(\psi_j\Gamma_\nu\nabla_\mu\xi^j\right)= &-\xi^1\left(\psi^2\slashed{\nabla}\xi^1\right)+\Gamma^\mu\Gamma^\nu\xi^1\left(\psi^2\Gamma_\nu\nabla_\mu\xi^1\right) \\
&-\xi^1\left(\psi^1\slashed{\nabla}\xi^2\right)+\Gamma^\nu\Gamma^\mu\xi^1\left(\psi^1\Gamma_\nu\nabla_\mu\xi^2\right).
\end{aligned}
\tag{B.18}
$$

Now we use 4D Fierz identities to simplify above terms. We have

$$
\begin{aligned}
-\xi^1\left(\psi^2\slashed{\nabla}\xi^1\right) &= -\tfrac{1}{2}\left(\slashed{\nabla}\xi^1\Gamma^\mu\xi^1\right)\Gamma_\mu\psi_2 = 2\beta\left(\xi^1\Gamma^\mu\xi^2\right)\Gamma_\mu\psi^2, \\
\Gamma^\mu\Gamma^\nu\xi^1\left(\psi^2\Gamma_\nu\nabla_\mu\xi^1\right) &= -2\Gamma^\mu\psi^2\left(\xi^1\nabla_\mu\xi^1\right)=-2\beta\left(\xi^1\Gamma^\mu\xi^2\right)\Gamma_\mu\psi^2.
\end{aligned}
\tag{B.19}
$$

So these two terms cancel each other. The next two terms become:

$$
\begin{aligned}
-\xi^1\left(\psi^1\slashed{\nabla}\xi^2\right) &= \tfrac{1}{2}\left(\slashed{\nabla}\xi^2\,\xi^1\right)\psi^1+\tfrac{1}{8}\left(\slashed{\nabla}\xi^2\,\Gamma^{\mu\nu}\xi^1\right)\Gamma_{\mu\nu}\psi^1=-\tfrac{\beta}{2}\left(\xi^1\Gamma^{\mu\nu}\xi^1\right)\Gamma_{\mu\nu}\psi^1, \\
+\Gamma^\nu\Gamma^\mu\xi^1\left(\psi^1\Gamma_\nu\nabla_\mu\xi^2\right) &= 2\psi^1\left(\nabla_\mu\xi^2\Gamma^\mu\xi^1\right)=8\beta\psi^1\left(\xi^1\xi^1\right)=0.
\end{aligned}
\tag{B.20}
$$

So,

$$
\Gamma^{\mu\nu}\xi^1\left(\psi_j\Gamma_\nu\nabla_\mu\xi^j\right)=-\tfrac{\beta}{2}\left(\xi^1\Gamma^{\mu\nu}\xi^1\right)\Gamma_{\mu\nu}\psi^1.
\tag{B.21}
$$

For CKS, the vector field $v^\mu \equiv \xi^j \Gamma^\mu \xi_j$ is a Killing vector field and it satisfies the following:

$$\nabla_\mu v_\nu = 2\beta \xi^1 \Gamma_{\mu\nu} \xi^1 + 2\beta \xi^2 \Gamma_{\mu\nu} \xi^2. \tag{B.22}$$

From this we can write

$$\nabla_\mu v_\nu \Gamma^{\mu\nu} \psi^1 = 2\beta \xi^1 \Gamma_{\mu\nu} \xi^1 \Gamma^{\mu\nu} \psi^1. \tag{B.23}$$

The term involving $\xi^2$ vanishes due to the first Fierz identity in eq. (A.11). Similar conclusions hold for $\delta^2 \psi^2$. Combining everything we conclude that for CKS in four dimensions

$$\delta^2 \psi^i = v^\mu \nabla_\mu \psi^i + \tfrac{1}{4} \nabla_\mu v_\nu \Gamma^{\mu\nu} \psi^i = \mathcal{L}_v \psi^i. \tag{B.24}$$

**6D**

Now we start wit the general form given in eq. (B.14). Using the second Fierz identity in eq. (A.13), we have

$$\begin{aligned}
-2\xi^1 \left( \xi^1 \slashed{\nabla} \psi^2 \right) &= \tfrac{1}{24} \left( \xi^1 \Gamma^{\mu\nu\rho} \xi^1 \right) \Gamma_{\mu\nu\rho} \slashed{\nabla} \psi^2 \\
&= -\tfrac{1}{24} \Gamma^\sigma \left[ \left( \xi^1 \Gamma^{\mu\nu\rho} \xi^1 \right) \Gamma_{\mu\nu\rho} \nabla_\sigma \psi^2 \right] + \tfrac{1}{4} \left( \xi^1 \Gamma^{\mu\nu\rho} \xi^1 \right) \Gamma_{\mu\nu} \nabla_\rho \psi^2 \\
&= \tfrac{1}{4} \left( \xi^1 \Gamma^{\mu\nu\rho} \xi^1 \right) \Gamma_{\mu\nu} \nabla_\rho \psi^2,
\end{aligned} \tag{B.25}$$

where the last equality follows due to the first Fierz identity in eq. (A.13).

Using the third identity in eq. (A.13), we get

$$\begin{aligned}
\Gamma^\nu \left( \Gamma^\mu \xi^1 \left( \xi^1 \Gamma_\nu \nabla_\mu \psi^2 \right) \right) &= \Gamma^\nu \left[ \tfrac{1}{4} \left( \xi^1 \Gamma^\mu \xi^1 \right) \Gamma_\nu \nabla_\mu \psi^2 - \tfrac{1}{8} \left( \xi^1 \Gamma^{\rho\sigma} \Gamma^\mu \xi^1 \right) \Gamma_{\rho\sigma} \Gamma_\nu \nabla_\mu \psi^2 \right] \\
&= -\tfrac{1}{4} \left( \xi^1 \Gamma^{\mu\nu\rho} \xi^1 \right) \Gamma_{\mu\nu} \nabla_\rho \psi^2,
\end{aligned} \tag{B.26}$$

where the second equality follows by using Clifford algebra commutation relations and noting that $\xi^1 \Gamma^\mu \xi^1 = 0$. Hence we see that the second and the third term in $\delta^2 \psi^1$ as given in B.14 cancel each other. Other terms appearing there can also be simplified in a similar fashion. For the fourth term, upon using Fierz identity and after some algebra we get

$$2\xi^2 \left( \xi^1 \slashed{\nabla} \psi^1 \right) = \tfrac{1}{2} v^\mu \Gamma_\mu \slashed{\nabla} \psi^1 - \tfrac{1}{4} \left( \xi^1 \Gamma^{\mu\nu\rho} \xi^2 \right) \Gamma_{\mu\nu} \nabla_\rho \psi^1. \tag{B.27}$$

And the last term in $\delta^2 \psi^1$ takes the following form after using Fierz identity and Clifford algebra commutation relations.

$$-\Gamma^\nu \left[ \Gamma^\mu \xi^1 \left( \xi^2 \Gamma_\nu \nabla_\mu \psi^1 \right) \right] = v^\mu \nabla_\mu \psi^1 + \tfrac{1}{4} \left( \xi^1 \Gamma^{\mu\nu\rho} \xi^2 \right) \Gamma_{\mu\nu} - \tfrac{1}{2} v^\mu \Gamma_\mu \slashed{\nabla} \psi^1. \tag{B.28}$$

Hence these two terms combine to give $v^\mu \nabla_\mu \psi^1$. One gets a similar result for $\psi^2$. So two supersymmetry variations of the guagino take the following form in both 4D and 6D:

$$\delta^2 \psi^i = v^\mu \nabla_\mu \psi^i + \Gamma^{\mu\nu} \xi^i \left( \psi_j \Gamma_\nu \nabla_\mu \xi^j \right). \tag{B.29}$$

For a KS, the second term vanishes and we get a Lie derivative along the Killing vector field $v^\mu$.

**(3) Auxiliary field**

For auxiliary fields, using covariantized supersymmetry transformations, one gets

$$\begin{aligned}
\delta^2 D^{ij} = &-\tfrac{1}{2} \left( \xi^i \Gamma^\rho \Gamma^{\mu\nu} \xi^j \right) \nabla_\rho F_{\mu\nu} - \tfrac{1}{2} \left( \xi^i \Gamma^\rho \Gamma^{\mu\nu} \nabla_\rho \xi^j \right) F_{\mu\nu} \\
&+ \left( \xi^i \Gamma^\mu \xi_k \right) \nabla_\mu D^{jk} + \left( \xi^i \slashed{\nabla} \xi_k \right) D^{jk} + (i \leftrightarrow j).
\end{aligned} \tag{B.30}$$

The first term in the above equation actually vanishes due to Bianchi identity and $i \leftrightarrow j$ anti-symmetry of $\xi^i \Gamma^\mu \xi^j$.

$$-\tfrac{1}{2}\left(\xi^i \Gamma^\rho \Gamma^{\mu\nu}\xi^j\right)\nabla_\rho F_{\mu\nu} + (i \leftrightarrow j) = -\left(\xi^i \Gamma^{\rho\mu\nu}\xi^j\right)\nabla_\rho F_{\mu\nu} - v_\mu \nabla_\nu F^{\nu\mu}\left(\epsilon^{ij}+\epsilon^{jI}\right) = 0. \quad \text{(B.31)}$$

The third term in $\delta^2 D^{ij}$ just becomes

$$\left(\xi^i \Gamma^\mu \xi_k\right)\nabla_\mu D^{jk} + (i \leftrightarrow j) = \frac{v^\mu}{2}\epsilon^i{}_k \nabla_\mu D^{jk} + \leftrightarrow J = v^\mu \nabla_\mu D^{ij}. \quad \text{(B.32)}$$

So the full action of two supersymmetry transformation on $D^{ij}$ becomes

$$\delta^2 D^{ij} = v^\mu \nabla_\mu D^{ij} + 2\left(\xi^{(i}\slashed{\nabla}\xi_k\right)D^{j)k} - \tfrac{1}{2}F_{\mu\nu}\left(\xi^i \Gamma^\rho \Gamma^{\mu\nu}\nabla_\rho \xi^j + i \leftrightarrow j\right). \quad \text{(B.33)}$$

In 4D, the last two terms vanish for all KS and CKS. In 6D, they vanish for KS but not for CKS.

## B.3 Two supersymmetry variations of the hypermultiplet

Here we give details of the computation of two supersymmetry transformations acting on fields in the hypermultiplet.

**Scalars**

We have

$$\begin{aligned}
\delta^2 \phi^{ij} &= \xi^i \delta \chi^{\bar{j}} = 2\xi^i \Gamma^\mu \xi_k D_\mu \phi^{k\bar{j}} + 8\xi^i \Gamma^\mu \xi_k \phi^{k\bar{j}}\partial_\mu f + \xi^i v_k K^{k\bar{j}} \\
&= v^\mu D_\mu \phi^{i\bar{j}} + 4v^\rho \partial_\rho f \phi^{i\bar{j}} + \left(\xi^i v_k\right)K^{k\bar{j}}.
\end{aligned} \quad \text{(B.34)}$$

**Fermions**

The two supersymmetry various of the fermion $\chi^{\bar{i}}$ take the following form

$$\begin{aligned}
\delta^2 \chi^{\bar{i}} = &\, 10\,\partial_\mu f \Gamma^\mu \xi_j \left(\xi^j \chi^{\bar{i}}\right) + 2\partial_\mu f \Gamma_\nu \xi_j \left(\xi^j \Gamma^{\mu\nu}\xi^{\bar{i}}\right) + 2\Gamma^\mu \xi_j \left(\xi^j D_\mu \chi^{\bar{i}}\right) + 2\Gamma^\mu \xi_j \left(\xi^k \Gamma_\mu \psi_k\right)\phi^{kI} \\
&- 2\left(v^j \slashed{D}\chi^{\bar{i}}\right)v_j - 4\left(v^j \psi_k\right)\phi^{k\bar{i}}v_j.
\end{aligned} \quad \text{(B.35)}$$

These terms can be brought in the desired form by using Fierz identities and relations satisfied by $v^i$s.

$$\begin{aligned}
10\,\partial_\mu f \Gamma^\mu \xi_j \left(\xi^j \chi^{\bar{i}}\right) &= \tfrac{5}{2}v^\mu \partial_\mu f \chi^{\bar{i}} - \tfrac{5}{2}v^\mu \partial^\nu f \Gamma_{\mu\nu}\chi^{\bar{i}}, \\
2\partial_\mu f \Gamma_\nu \xi_j \left(\xi^j \Gamma^{\mu\nu}\xi^{\bar{i}}\right) &= \tfrac{5}{2}v^\mu \partial_\mu f \chi^{\bar{i}} + \tfrac{3}{2}v^\mu \partial^\nu f \Gamma_{\mu\nu}\chi^{\bar{i}}, \\
2\Gamma^\mu \xi_j \left(\xi^j D_\mu \chi^{\bar{i}}\right) &= \tfrac{1}{2}v^\mu D_\mu \chi^{\bar{i}} - \tfrac{1}{2}v_\mu \Gamma^{\mu\nu}D_\nu \chi^{\bar{i}}, \\
2\Gamma^\mu \xi_j \left(\xi^k \Gamma_\mu \psi_k\right)\phi^{kI} &= -v^\mu \Gamma_\mu \psi_j \phi^{j\bar{i}}, \\
-2\left(v^j \slashed{D}\chi^{\bar{i}}\right)v_j &= \tfrac{1}{2}v^\mu D_\mu \chi^{\bar{i}} + \tfrac{1}{2}v^\mu \Gamma_{\mu\nu}D_\nu \chi^{\bar{i}}, \\
-4\left(v^j \psi_k\right)\phi^{k\bar{i}}v_j &= v^\mu \Gamma_\mu \psi_j \phi^{j\bar{i}}.
\end{aligned} \quad \text{(B.36)}$$

Combining all these we get

$$\delta^2 \chi^i = \mathcal{L}_v \chi^{\bar{i}} + 5v^\mu \partial_\mu f \chi^{\bar{i}}. \quad \text{(B.37)}$$

**Auxiliary fields**

We have

$$\delta^2 K^{i\bar{j}} = -2\nu^i \delta \left( \slashed{D} \chi^{\bar{j}} \right) - 4\nu^i \delta \left( \psi_k \phi^{k\bar{j}} \right), \tag{B.38}$$

After using the supersymmetry transformation of $\chi^{\bar{i}}$ and the vector field $A_\mu$, the first term in above equation becomes

$$-2\nu^i \delta \left( \slashed{D} \chi^{\bar{j}} \right) = \nu^\mu D_\mu K^{i\bar{j}} + 6\nu^\mu \partial_\mu f K^{i\bar{j}} - 2 \left( \nu^i \Gamma^{\mu\nu} \xi_k \right) F_{\mu\nu} \phi^{k\bar{j}} \\ - 4 \left( \nu^i \xi_k \right) \left( D^2 \phi^{k\bar{j}} - \frac{6}{r^2} \phi^{k\bar{j}} \right) + 2 \left( \nu^i \Gamma^\mu \chi^{\bar{j}} \right) \left( \xi^k \Gamma_\mu \psi_k \right). \tag{B.39}$$

Last term can be modified using Fierz identities resulting into

$$-2\nu^i \delta \left( \slashed{D} \chi^{\bar{j}} \right) = \nu^\mu D_\mu \psi^{i\bar{j}} + 6\nu^\mu \partial_\mu f K^{i\bar{j}} - 2 \left( \nu^i \Gamma^{\mu\nu} \xi_k \right) F_{\mu\nu} \phi^{k\bar{j}} \\ - 4 \left( \nu^i \xi_k \right) \left( D^2 \phi^{k\bar{j}} - \frac{6}{r^2} \phi^{k\bar{j}} + \frac{3}{4} \psi^k \chi^{\bar{j}} \right) - \frac{1}{2} \left( \nu^i \Gamma^{\mu\nu} \xi^k \right) \left( \psi_k \Gamma_{\mu\nu} \chi^{\bar{j}} \right). \tag{B.40}$$

Second term in $\delta^2 K^{i\bar{j}}$ takes the form

$$4\nu^i \delta \left( \psi_k \phi^{k\bar{j}} \right) = 2 \left( \nu^i \Gamma^{\mu\nu} \xi_k \right) F_{\mu\nu} \phi^{k\bar{j}} - 4 \left( \nu^i \xi_k \right) D_j^{\ k} \phi^{j\bar{j}} + 4 \left( \nu^i \psi_j \right) \left( \xi^j \chi^{\bar{j}} \right). \tag{B.41}$$

After using Fierz identities to simplify the last term, we get

$$4\nu^i \delta \left( \psi_k \phi^{k\bar{j}} \right) = \ 2 \left( \nu^i \Gamma^{\mu\nu} \xi_k \right) F_{\mu\nu} \phi^{k\bar{j}} - 4 \left( \nu^i \xi_k \right) \left( D_j^{\ k} \phi^{j\bar{j}} + \frac{1}{4} \psi^k \chi^{\bar{j}} \right) \\ + \frac{1}{2} \left( \nu^i \Gamma^{\mu\nu} \xi^k \right) \left( \psi_k \Gamma_{\mu\nu} \chi^{\bar{j}} \right). \tag{B.42}$$

Combining all the term we finally get

$$\delta^2 K^{i\bar{j}} = \mathcal{L}_\nu K^{i\bar{j}} + 2\Omega_Y \nu^\mu \partial_\mu f K^{i\bar{j}} - 4 \left( \nu^i \xi_j \right) \left( D^2 \phi^{j\bar{j}} - \frac{6}{r^2} \phi^{j\bar{j}} + D_k^{\ j} \phi^{k\bar{j}} + \psi^j \chi^{\bar{j}} \right). \tag{B.43}$$

## B.4   Almost complex structures

Here we prove the identities of eq. (5.7) for $\mathcal{I}_{\mu\nu} \equiv i\epsilon^\dagger \Gamma_{\mu\nu} \epsilon$, where $\epsilon$ is a constant spinor. Using Fierz identity on the third line of A.13 we can write:

$$\mathcal{I}_\mu^{\ \rho} \mathcal{I}_\rho^{\ \nu} = -\frac{1}{4} \epsilon^\dagger \Gamma_{\mu\rho} \Gamma^{\rho\nu} \epsilon + \frac{1}{8} \epsilon^\dagger \Gamma_{\mu\rho} \Gamma^{\sigma\delta} \Gamma^{\rho\nu} \epsilon \epsilon^\dagger \Gamma_{\sigma\delta} \epsilon \\ = \delta_\mu^{\ \nu} \left( -\frac{5}{4} + \frac{1}{8} \epsilon^\dagger \Gamma^{\sigma\delta} \epsilon \epsilon^\dagger \Gamma_{\sigma\delta} \epsilon \right) - \mathcal{I}_\mu^{\ \rho} \mathcal{I}_\rho^{\ \nu}. \tag{B.44}$$

The second equality is obtained by using standard gamma matrix relations and the definition of $\mathcal{I}_\mu^{\ \nu}$. The same Fierz identity also gives $\epsilon^\dagger \Gamma^{\sigma\delta} \epsilon \epsilon^\dagger \Gamma_{\sigma\delta} \epsilon = -6$. Using this on easily obtains

$$\mathcal{I}_\mu^{\ \rho} \mathcal{I}_\rho^{\ \nu} = -\delta_\mu^{\ \nu}. \tag{B.45}$$

To prove the second identity we start by using eq. (A.8) to write

$$\varepsilon^{\mu_1 \mu_2 \mu_3 \mu_4 \mu_5 \mu_6} \mathcal{I}_{\mu_3 \mu_4} \mathcal{I}_{\mu_5 \mu_6} = 2i\epsilon^\dagger \Gamma_{\mu_3 \mu_4} \epsilon \epsilon^\dagger \Gamma^{\mu_1 \mu_2 \mu_3 \mu_4} \epsilon \\ = 2i\epsilon^\dagger \Gamma_{\mu_3 \mu_4} \epsilon \epsilon^\dagger \Gamma^{\mu_3 \mu_4} \Gamma^{\mu_1 \mu_2} \epsilon + 4i\epsilon^\dagger \Gamma^{\mu_1 \mu_2} \epsilon \\ = -8i\epsilon^\dagger \Gamma^{\mu_1 \mu_2} \epsilon = -8\mathcal{I}^{\mu_1 \mu_2}, \tag{B.46}$$

where the second equality follows by expressing the rank four gamma matrix in terms of lower rank matrices. The third equality follows by using Fierz identity.

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
