# Peer review of "(1,0) gauge theories on the six-sphere"

_SciPost Physics, doi:SciPost Phys. 6, 002 (2019)_

## Round 2 · Referee Report · Anonymous (Referee 1) · 2018-11-14

Strengths

1- constructs a 6D minimal SUSY gauge theory on the 6-sphere 2- discusses SUSY localization

Weaknesses

Inclusion of tensor multiplets as well as the evaluation of SUSY observables are left as future problems

Report

It was shown that the gauge coupling has to be made coordinate-dependent, which looks a peculiar feature in 6D. The localization locus for vectormultiplet is shown to be the solutions of the Hermitian Yang-Mills equation. An intriguing condition is also derived for hypermultiplet.

Requested changes

1- in section 2, the action and transformation rule (2.1) and (2.2) look as if the number of fermionic DOF in a single multiplet is twice the correct number. It appears some chirality condition for the fermions $\psi^1,\psi^2$ are needed. The paper claims consistency with the ref [12], but in that reference the chirality projection is imposed after (A.23).

2- A more minor point is that in the second equation in (5.2) the RHS is just the bosonic part of the LHS and therefore not precisely equal to the LHS.

  • validity: good
  • significance: ok
  • originality: good
  • clarity: good
  • formatting: good
  • grammar: good

Author:  Usman Naseer  on 2018-11-25  [id 355]

(in reply to Report 1 on 2018-11-14)
Category:
remark

I thank the referee for a careful reading of the manuscript and her/his comments. Regarding changes requested: 1- There is indeed a chirality condition but I failed to mention it explicitly in section 2. I do mention the chirality conditions later on in section 3 but I should clearly state it in section 2. 2- Thank you for pointing out this oversight. I should fix that in the final version.

Thank you, -Usman

---

## Round 2 · Referee Report · Benjamin Assel (Referee 2) · 2018-11-22

Strengths

1- Computations are given in full details 2- The discussion is very clear and self-contained 3- The results are new

Weaknesses

1- The motivation for the construction is towards localization computations, however, in this specific context, such a computation seems beyond the reach of current techniques in the field.

Report

In this paper the author constructs the action and supersymmetry transformations on the round six-sphere for the vector multiplet and hypermultiplet of a gauge theory with minimal $\mathcal{N}=(1,0)$ supersymmetry. The construction preserves 16 supersymmetries on-shell and an extension is given which preserves one supersymmetry off-shell (for the hypermultiplet in particular). The remarkable feature of the construction is the presence of a space dependent dilaton factor in front of the vector multiplet action, which makes the construction possible. A first step towards computing the sphere partition function of the theory using supersymmetric localization is taken, by showing that the localization locus comprises gauge fields satisfying Hermitian Yang-Mills equations (the 6d analogue of self-dual equations in 4d) and hypermultiplet scalars satisfying some mixed holomorphic condition. These conditions involve an almost complex structure defined from the Killing spinor used in the localization argument.

The discussion and computations are very detailed and transparent, showing a work of very good quality.

I am still puzzled by one point. In this construction, the sphere Lagrangian has the overall factor $e^\phi = (1+\beta^2x^2)^2$ which diverges at one pole of the sphere ($x \to \infty$). This might make the action diverge for a generic field configuration on the sphere. It seems that the pole of the sphere might be seen as a boundary where one would need to impose boundary conditions on the fields to make the action finite. That could have important consequences on a hypothetical localization computation.

Finally I noticed some probable typos. The definition of $F^\pm$ in (5.9) seems to be missing some terms. The asymmetry between the expressions for $F^+$ and $F^-$ in (5.10) might also be a misprint.

Requested changes

1- Correct misprints.

  • validity: top
  • significance: good
  • originality: ok
  • clarity: top
  • formatting: perfect
  • grammar: excellent

Author:  Usman Naseer  on 2018-11-25  [id 356]

(in reply to Report 2 by Benjamin Assel on 2018-11-22)
Category:
remark

Dear Benjamin,

Thank you very much for a careful reading of the manuscript and insightful comments.

Regarding the point that you are puzzled about: One can do a field redefinition to write Lagrangian with the "canonical" coupling. In this case, it corresponds to letting $A_\mu\to (1+\beta^2 x^2)^{-1} A_\mu$ (and similarly for fermion). This will get rid of the overall factor and all the terms in the Lagrangian would then be manifestly regular everywhere on the sphere. I am aware of this point and I address it in an upcoming paper.

Re. equations (5.9) and (5.10): Eq. (5.9) indeed has a typo. $F^\pm$ satisfy the property that $F^{\pm}=\pm \star \left( F^{\pm} \wedge {\cal I}\right)$---thank you for pointing this out. The asymmetry in eq. (5.10), however, is correct as it is.

Thank you,
-Usman

Benjamin Assel  on 2018-11-26  [id 357]

(in reply to Usman Naseer on 2018-11-25 [id 356])
Category:
validation or rederivation

I think this point on the field redefinition and regularity of the resulting terms should be mentioned in the paper.
Otherwise I recommend the paper for publication.

---

## Round 2 · Referee Report · Anonymous (Referee 3) · 2018-11-25

Strengths

1- The computation is clearly explained and many details are provided. This can be useful to future work on this topic. 2- Supersymmetric localization in 6d is still in its infancy. This is a welcome stimulant for further thinking.

Weaknesses

1- The work is mainly of technical nature. 2- It is limited to gauge fields and hypers, as an obvious generalization to what has been done in lower dimensions. A more thorough discussion of N=(1,0) supersymmetry on S^6 would have included the tensor multiplet.

Report

The author discusses 6d N=(1,0) supersymmetric gauge theories on the six-sphere ($S^6$) with its round metric. By a direct computation, it is shown that one can preserve supersymmetry only at the expense of introducing a non-trivial dilaton coupling, which can be interpreted as a space-dependent gauge coupling. The $S^6$ is viewed as an $S^5$ fibered over an interval, and the 6d gauge coupling goes from zero to a finite value at the end points of the interval.

The author then considers briefly the localization locus, using a standard $L=|Q\psi|^2$ localization term. It is shown that the vector multiplet localizes onto the solutions to the 6d Hermitian-Yang-Mills equation. A localization equation is derived for the hyper as well.

Except for the introduction of a dilaton, the computation is standard and appears to be correct. A lot of details are provided about the supersymmetry variations (maybe even too much). A more thorough discussions of the localization equations is left for future work.

My main question about this work is the significance of the divergence of the effective $1/g^2$ at the south pole. That divergence suggests that one might actually be considering a non-compact space, $\mathbb{R}^6$ with a non-trivial metric, instead of the actual six-sphere, since the supersymmetric YM action for a generic 6d connection will be non-zero only if some boundary conditions are chosen (such as possibly $A_\mu=0$ at the south pole). On the other hand, 6d gauge fields are only low-energy effective descriptions in 6d, and the correct interpretation of this divergence might be related to the UV completion.

One can formulate that question another way: if the claim is that we have an off-shell realization of supersymmetry on $S^6$, how could the gauge coupling be space-dependent? What probably happens is that there must be some universal dilaton coupling. The supersymmetric background should be understood as some background supergravity, à la Festuccia-Seiberg. Where does this dilaton appears in such a picture?

Finally, here are a few typos as well as suggestions:
-"vector-multiplet" should be spelled "vector multiplet" if it is not used as an adjective.
-in the abstract and elsewhere, one refers to "the field strength of the scalar" in the hypermultiplet. That seems to be a non-standard terminology. Maybe "covariant derivative of the scalar" would be more appropriate.
-end of intro: "furhter"->"further"
-in sec.2, the author might want to cite [arXiv:1209.5408], which also discussed 4d N=1 susy on $S^4$.
-before eq.(4.6): "cupling"-->"coupling"
-after eq.(4.12): "in above arguments"-->"in the above argument"
-intro to sec.5: "except the south pole"-->"except at the south pole"
-before eq.(5.3): "contirbute"-->"contribute"
-middle of sec.6: "be achieve by"-->"be achieved by"
-end of sec.6: "à la localization" with "à"

Requested changes

1-fix the above typos.

  • validity: good
  • significance: ok
  • originality: ok
  • clarity: good
  • formatting: good
  • grammar: good

Author:  Usman Naseer  on 2018-11-27  [id 360]

(in reply to Report 3 on 2018-11-25)
Category:
answer to question
suggestion for further work

I thank the referee for her/his comments and reading the draft very carefully as is evident from numerous spotted typos. Here are my thoughts about some of the comments that the referee has made:

1) Re. (1,0) tensor multiplet: I agree that inclusion of (1,0) tensor multiplet is an important and interesting future direction. Off-shell formulation of tensor multiplet, even on flat space, is not well understood so this presents a challenge as it is not clear what should be the starting point to put the theory on sphere. That said, I hope to come back to this issue in near future.

2) Re. significance of the divergence of effective $1/g^2$: This point was raised by another referee as well. The "apparent" divergence is an artifact of the choice of the field variables we make. Namely one can choose to write the Lagrangian in terms of "canonical" coupling and fields. This just corresponds to letting $A_\mu\to g(x) A_\mu$, as usual. One can then see that all terms appearing in the Lagrangian are regular everywhere on the sphere.

3) Re. SUSY background: This is one good way to understand my results. Indeed my investigation was motivated by the fact that, in principle, one can have supersymmetric backgrounds with non-trivial values of metric as well as other fields in the gravity multiplet and that will lead to supersymmetric theories with space dependent couplings, mass parameters etc. It will be interesting to derive my results as well as other 6D supersymmetric backgrounds from an off-shell supergravity à la Festuccia-Seiberg

Thank you, -Usman

---

## Round 3 · Referee Report · Anonymous (Referee 3) · 2018-12-5

Strengths

See previous version.

Weaknesses

See previous version.

Report

This is an interesting and topical paper. See my review of the previous version. The new version has only minor changes. This is a good paper that I would recommend for publication.

Requested changes

none.

---

## Round 3 · Referee Report · Benjamin Assel (Referee 2) · 2018-12-5

Report

The modifications address the minor issues raised in the previous report.

---

## Round 3 · Author Response

Dear Editor,

In this revised version I have fixed several typos pointed out by all three referees. An important change is the discussion about the regularity of the vector-multiplet action at the end of section 3. This point was raised by two referees.

Thank you,
-Usman

---

## Round 3 · List of Changes

1. "vector-multiplet" changed to "vector multiplet" in a lot of places.
  2. end of intro: "furhter"->"further"
  3. in sec.2, new reference is added [arXiv:1209.5408], which also discussed 4d N=1 susy on S^4. 4.before eq.(4.6): "cupling"-->"coupling" 5.after eq.(4.12): "in above arguments"-->"in the above argument" 6.intro to sec.5: "except the south pole"-->"except at the south pole" 7.before eq.(5.3): "contirbute"-->"contribute" 8.middle of sec.6: "be achieve by"-->"be achieved by" 9.end of sec.6: "à la localization" with "à".
  4. Typo in eq. 5.9 is fixed.
  5. At the end of section 3, a field redefinition is discussed which makes the Lagrangian manifestly regular everywhere.
  6. chirality condition is explicitly mentioned in the beginning of section 2 after 2.2

---

## Editorial Decision

published